# Hindsight Preference Learning for Offline Preference-based Reinforcement Learning

## Abstract

Offline preference-based reinforcement learning (RL), which focuses on optimizing policies using human preferences between pairs of trajectory segments selected from an offline dataset, has emerged as a practical avenue for RL applications. Existing works rely on extracting *step-wise* reward signals from *trajectory-wise* preference annotations, assuming that preferences correlate with the cumulative Markovian rewards. However, such methods fail to capture the holistic perspective of data annotation: Humans often assess the desirability of a sequence of actions by considering the overall outcome rather than the immediate rewards. To address this challenge, we propose to model human preferences using rewards conditioned on future outcomes of the trajectory segments, i.e. the *hindsight information*. For downstream RL optimization, the reward of each step is calculated by marginalizing over possible future outcomes, the distribution of which is approximated by a variational auto-encoder trained using the offline dataset. Our proposed method, *Hindsight Preference Learning (HPL)*, can facilitate credit assignment by taking full advantage of vast trajectory data available in massive unlabeled datasets. Comprehensive empirical studies demonstrate the benefits of HPL in delivering robust and advantageous rewards across various domains.

## 1 Introduction

Although reinforcement learning (RL) has demonstrated remarkable success in various sequential decision-making tasks (Vinyals et al., 2019; Ye et al., 2020), its application in real-world scenarios remains challenging for practitioners, primarily due to two key reasons. First, modern RL methods typically require millions of online interactions with the environment (Haarnoja et al., 2018), which is prohibitively expensive and potentially dangerous in embodied applications (Levine et al., 2020). Second, reward engineering is necessary to align the induced behavior of the policy with human interests (Gupta et al., 2022; Knox et al., 2023). However, tweaking the reward requires substantial effort and extensive task knowledge of real-world scenarios. Reward hacking frequently occurs when the reward is improperly configured, leading to unintended consequences (Skalse et al., 2022).

There are several research directions aiming for addressing above challenges (Knox & Stone, 2009; Ng & Russell, 2000; Sadigh et al., 2017), among which *offline Preference-based RL (offline PbRL)* has gained increasing attention recently (Hejna & Sadigh, 2023; An et al., 2023; Kang et al., 2023). In offline PbRL, an offline dataset is collected by deploying a behavior policy, after which human labelers are required to provide relative preferences between two trajectory segments selected from the offline dataset. Offline PbRL significantly reduces the burden on human effort given that labeling preferences between trajectories is considerably easier compared to crafting step-wise reward signals. It has proven effective in large-scale applications including fine-tuning large language models (Touvron et al., 2023; Rafailov et al., 2023; Hu et al., 2023a).

Offline PbRL typically follows a two-phase paradigm: 1) learning a reward function that aligns with human preferences using a small labeled preference dataset; 2) applying the reward function to label a massive unlabeled dataset and performing policy optimization (Christiano et al., 2017). For the first phase, existing methods employ Bradley-Terry model (Bradley & Terry, 1952) to learn *step-wise* reward signals from *trajectory-wise* preferences based on the *Markovian reward assumption*: the preference correlates with the cumulative rewards of each trajectory (Christiano et al., 2017). However, as previous works (Kim et al., 2023) have unveiled, such an assumption is technically

Figure 1: Illustration of the reward learning procedure in HPL. Unlike previous methods, HPL first generates embeddings $z_t$ to encode the future part of the segments and optimize a reward function $r_\psi$ which is conditioned on the $s_t$, $a_t$ and the future $z_t$ using the Bradley-Terry model.

flawed and limited since humans evaluate the trajectory segments from a global perspective, making the obtained reward bear an implicit dependence on the future part of the segment. Consider the case of purchasing lottery tickets as an example. Although the expected return is negative, the payoff can be substantial if one wins. However, when assigning credits, we should allocate higher rewards to purchasing tickets only when we are certain of winning, rather than unconditionally encouraging such behavior.

In light of this, we propose *Hindsight Preference Learning (HPL)* to account for such dependence on future information. The key idea of HPL is to develop a *hindsight preference model*, which models human preferences using a reward function conditioning on the state $s$, action $a$ and the future trajectory starting from $(s, a)$, i.e. the *hindsight information*. In particular, given a $H$-length trajectory $\sigma_{1:H} = (s_1, a_2, s_2, a_2, \ldots, s_H, a_H)$, the reward of $s_t, a_t$ for $1 \le t \le H$ is given by $r(s_t, a_t | \sigma_{t:t+k})$. When labeling the unlabeled dataset, a scalar reward signal is computed for each state-action pair by marginalizing over all possible hindsight information, $r(s, a) = \int_\sigma p(\sigma | s, a) r(s, a | \sigma) d\sigma$. To deal with the high-dimensional nature of hindsight information, we pre-train a variational auto-encoder to efficiently represent the hindsight information, making the above marginalization feasible in practice. The reward learning procedure of HPL is illustrated in Figure 1.

HPL has two key benefits over prior works. First, by leveraging hindsight information in preference modeling, it captures the implicit holistic perspective of human preference labeling, addressing the key limitation of the Markovian reward assumption adopted in previous works. Second, the two-phase paradigm of offline PbRL might become less effective if there is a substantial distribution mismatch between preference and unlabeled dataset. HPL can take advantage of the unlabeled dataset by learning a prior over future outcomes. This allows for incorporating the information of trajectory distribution carried by the unlabeled dataset, thus delivering robust and advantageous rewards when labeling the unlabeled dataset. We provide extensive evaluations to verify these benefits.

## 2 PRELIMINARIES

### 2.1 MARKOV DECISION PROCESS

In standard RL, an agent interacts with an environment characterized by a Markov Decision Process (MDP) $\langle \mathcal{S}, \mathcal{A}, r, T, \gamma \rangle$ according to its policy $\pi(a|s)$. Here, $\mathcal{S}$ and $\mathcal{A}$ represent the state space and action space respectively, while $r(s, a)$ denotes the reward function, $T(s'|s, a)$ is the transition function, and $\gamma$ is the discounting factor. The value function defines the expected cumulative reward by following the policy $\pi(a|s)$,

$$V^\pi(s) = \mathbb{E}_{a_t \sim \pi(\cdot|s_t), s_{t+1} \sim T(\cdot|s_t, a_t)} \left[ \sum_{t=0}^\infty \gamma^t r(s_t, a_t) | s_0 = s \right]. \tag{1}$$

The primary objective of RL algorithms is to find an optimal policy that maximizes $\mathbb{E}_{s_0 \sim \mu_0}[V^\pi(s_0)]$, where $\mu_0$ is the initial state distribution.

## 2.2 OFFLINE PREFERENCE-BASED REINFORCEMENT LEARNING

In this work, we consider the problem of learning an optimal decision-making policy from a previously collected dataset with *preference feedback*. In its generalist framework, the ground-truth reward is not given in the data. Instead, the learner is provided with a *preference dataset* and a massive *unlabeled dataset*, and follows a two-phase paradigm: 1) *reward learning*, learn a reward model $r$ with the preference data; and 2) *reward labeling*, apply $r$ to label the unlabeled dataset in order to perform policy optimization with a large amount of data.

Let $\sigma = (s_1, a_1, s_2, a_2, \ldots, s_{|\sigma|}, a_{|\sigma|})$ denote a consecutive segment of states and actions from a trajectory. The preference dataset $\mathcal{D}_{\mathrm{p}} = \{(\sigma_i^0, \sigma_i^1, y_i)\}_{i=1}^{|\mathcal{D}_{\mathrm{p}}|}$ consists of segments pairs with preference label given by the human annotators. The preference label is given by: $y_i = 0$ if $\sigma^0 \succ \sigma^1$ and $y_i = 1$ if $\sigma^1 \succ \sigma^0$, where we use $\sigma \succ \sigma'$ to denote $\sigma$ is more preferred than $\sigma'$. The unlabeled dataset $\mathcal{D}_{\mathrm{u}}$ contains reward-free trajectories $\{\sigma_i\}_{i=1}^{|\mathcal{D}_{\mathrm{u}}|}$ collected by some behavior policy $\pi_\beta$. In practice, we usually have $|\mathcal{D}_{\mathrm{u}}| \gg |\mathcal{D}_{\mathrm{p}}|$ as collecting human annotations is more time-consuming and expensive compared to collecting unlabeled trajectories.

To learn a reward function $r$, a common approach is to assume a probabilistic preference model $P$ and maximize the likelihood of the preference dataset,

$$\mathcal{L}(\psi) = -\sum_{(\sigma^0, \sigma^1, y) \in \mathcal{D}_{\mathrm{p}}} (1-y) \log P(\sigma^0 \succ \sigma^1; \psi) + y \log P(\sigma^1 \succ \sigma^0; \psi), \qquad (2)$$

where $P(\sigma \succ \sigma'; \psi)$ is the preference model parameterized by the parameters $\psi$. For the probabilistic preference model, most existing methods adopt the Markovian reward assumption (Christiano et al., 2017; Shin et al., 2023; Hu et al., 2023b):

$$\rho_{\mathrm{MR}}(\sigma; \psi) = \sum_{(s,a) \in \sigma} r_\psi(s, a). \qquad (3)$$

That is, the *preference strength* of a segment $\sigma$ correlates with its cumulative rewards. Applying the Bradley-Terry model (Bradley & Terry, 1952) leads to the following *Markovian Reward (MR)* preference model,

$$P_{\mathrm{MR}}(\sigma^0 \succ \sigma^1; \psi) = \frac{\exp(\rho_{\mathrm{MR}}(\sigma^0; \psi))}{\exp(\rho_{\mathrm{MR}}(\sigma^0; \psi)) + \exp(\rho_{\mathrm{MR}}(\sigma^1; \psi))}. \qquad (4)$$

Plugging Equation 4 into Equation 2 yields a practical learning objective for learning the reward function. Finally, we can label $\mathcal{D}_{\mathrm{u}}$ with the learned reward model for any $(s, a) \in \sigma, \sigma \in \mathcal{D}_{\mathrm{u}}$. The resulting labeled dataset can be used for policy optimization with any offline RL algorithms, such as IQL (Kostrikov et al., 2022) and AWAC (Nair et al., 2020).

## 3 HINDSIGHT PREFERENCE LEARNING

In this section we introduce a new preference model designed to address the limitations of the MR preference model by utilizing hindsight information. We begin with an illustrative example that serves as the primary motivation for our approach, followed by a detailed explanation of the formalization and implementation of the proposed method.

### 3.1 MOTIVATING EXAMPLE: THE INFLUENCE OF THE FUTURE

To elucidate the influence of the future in preference modeling, we take the gambling MDP from Yang et al. (2023) as an example (Figure 2). An agent begins at $s_1$ with two actions: $a_1$ and $a_2$. Choosing the high-risk action $a_1$, the agent transitions to a rewarding state $s_{\mathrm{good}}$ with probability of $10\%$, but is more likely ($90\%$) to a penalizing state $s_{\mathrm{bad}}$. Alternatively, the safer and actually optimal action $a_2$ consistently leads to a neutral state $s_{\mathrm{avg}}$, yielding a reward of 0. Suppose we are to extract the reward function using the provided dataset, where preferences are labeled based on the ground-truth reward:

$$\mathcal{D} = \begin{cases} ((s_1 \to a_1 \to s_{\mathrm{good}} \to a_3), & (s_1 \to a_2 \to s_{\mathrm{avg}} \to a_3), & y = 0) \\ ((s_1 \to a_1 \to s_{\mathrm{good}} \to a_3), & (s_1 \to a_2 \to s_{\mathrm{avg}} \to a_3), & y = 0) \\ ((s_1 \to a_1 \to s_{\mathrm{good}} \to a_3), & (s_1 \to a_1 \to s_{\mathrm{bad}} \to a_3), & y = 0) \\ ((s_1 \to a_1 \to s_{\mathrm{bad}} \to a_3), & (s_1 \to a_2 \to s_{\mathrm{avg}} \to a_3), & y = 1) \end{cases}.$$

Applying the MR preference model (Equation 4) to this dataset would likely yield a reward function where $r_\psi(s_1, a_1) > r_\psi(s_1, a_2)$, because a larger proportion of trajectories involving $a_1$ lead to the rewarding state $s_{\text{good}}$ (our experiments also validate this in Figure 3). However, this violates rationality as selecting $a_2$ offers a higher return in expectation. This failure can be attributed to the inappropriate credit assignment inherent in the MR preference model (Equation 4): a preference for $(s_1 \to a_1 \to s_{\text{good}} \to a_3)$ will assign credits to both $r_\psi(s_1, a_1)$ and $r_\psi(s_{\text{good}}, a_3)$ equally, leading to over-estimated rewards for $a_1$. To address this issue, a natural approach is to condition the reward of $(s_1, a_1)$ on the future outcome of the segment (i.e. $s_{\text{good}}$ or $s_{\text{bad}}$), so that $a_1$ is encouraged only when it leads to a favorable outcome $s_{\text{good}}$. The conditional reward function $r_\psi(s, a|\sigma_{\text{future}})$ thus answers the critical question: *If my future is determined to be $\sigma_{\text{future}}$, how advantageous it is for me to choose action $a$ at $s$?*

When applying $r_\psi(s, a|\sigma_{\text{future}})$ to label data, we can marginalize over all possible future segments according to some prior distribution $p_{\text{prior}}(\cdot|s, a)$ to get the value $r_\psi(s, a) = \mathbb{E}_{\sigma \sim p_{\text{prior}}(\cdot|s,a)}[r_\psi(s, a|\sigma)]$. Note that the prior distribution can be estimated using the unlabeled offline dataset, which is unbiased concerning the environment transition and the behavior policy. Take the gambling MDP again as an example, the marginalization would decrease the reward of action $a_1$ since most of the time (90%) the agent will arrive at the bad state. Besides, when there exists a mismatch between the data distribution of the preference dataset $\mathcal{D}_{\text{p}}$ and that of the unlabeled dataset $\mathcal{D}_{\text{u}}$ (which is a common

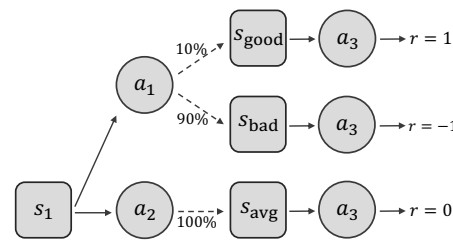

Figure 2: A gambling MDP that illustrates the potential failure modes of the MR preference model.

occurrence because of the improvement of the policy or the non-uniform sampling for preference pairs), it is found that the reward from existing methods may not align with the RL agent's interests (Hu et al., 2023b; Liu et al., 2023). The conditional reward model provides an effective solution to this issue by enabling the marginalization of the reward function $r_\psi$ using a data distribution that is more closely aligned with the on-policy data, thereby improving the alignment between learned behaviors and desired outcomes.

## 3.2 HINDSIGHT PREFERENCE MODEL

We now present *Hindsight Preference Model (HPM)*, a novel preference model that incorporates future information in preference modeling. As opposed to Equation 3, HPM assumes that the preference strength of a trajectory segment $\sigma = (s_0, a_0, s_1, a_1, \ldots, s_H)$ is determined by

$$\rho_{\text{HPM}}(\sigma; \psi) = \sum_{(s_t, a_t) \in \sigma} r_\psi(s_t, a_t | \sigma_{t:t+k}), \tag{5}$$

where $\sigma_{i:j}$ denotes the subsequence of $\sigma$ between steps $i$ and $j$.[1] That is, in HPM the reward function $r_\psi$ not only takes the current state $s_t$ and action $a_t$ as input, but also depends on the $k$-step future outcome $\sigma_{t:t+k}$. Then given a segment pair $(\sigma^0, \sigma^1)$, HPM models their preference by

$$P_{\text{HPM}}(\sigma^0 \succ \sigma^1; \psi) = \frac{\exp(\rho_{\text{HPM}}(\sigma^0; \psi))}{\exp(\rho_{\text{HPM}}(\sigma^0; \psi)) + \exp(\rho_{\text{HPM}}(\sigma^1; \psi))}. \tag{6}$$

In practice, directly implementing this conditional reward $r_\psi(s_t, a_t | \sigma_{t:t+k})$ is challenging due to the high-dimensional nature of the $k$-step segment $\sigma_{t:t+k}$. We address this issue by compressing the segment into a compact embedding rather than operating directly in the raw space of trajectory.

## 3.3 FUTURE SEGMENT EMBEDDING

We propose to compress future segments $\sigma_{t:t+k}$ into a compact embedding vector $z_t$ by training a conditional *Variational Auto-Encoder* (VAE) (Kingma & Welling, 2013). The architecture of our

---

[1]We use $\sigma_{t:t+k}$ to simplify the notations. For $t + k > H$, the subsequence will be clipped to $\sigma_{t:\min(t+k,H)}$.

---

**Algorithm 1** Hindsight Preference Learning (HPL)

---

**Input**: Preference dataset $\mathcal{D}_{\mathrm{p}} = \{(\sigma_i^0, \sigma_i^1, y_i)\}_{i=1}^{|\mathcal{D}_{\mathrm{p}}|}$, unlabeled dataset $\mathcal{D}_u = \{\sigma_i\}_{i=1}^{|\mathcal{D}_{\mathrm{u}}|}$

1: // VAE Training
2: Initialize the segment encoder $q_\theta$, decoder $p_\theta$, and prior $f_\theta$ with parameters $\theta$
3: **for** $n = 1, 2, \ldots, N_{\mathrm{VAE}}$ **do**
4:   Sample minibatch of segments $\sigma \sim \mathcal{D}_u$
5:   Update $\theta$ by maximizing Equation 7
6: **end for**
7: // Reward Learning
8: Initialize the reward function $r_\psi$ with parameters $\psi$
9: **for** $n = 1, 2, \ldots, N_{\mathrm{HPM}}$ **do**
10:   Sample minibatch of preference pairs $(\sigma^0, \sigma^1, y) \sim \mathcal{D}_p$
11:   Update $\psi$ by minimizing Equation 2 with $P_{\mathrm{HPM}}$ defined in Equation 8
12: **end for**
13: // RL Training
14: Label the reward for $\mathcal{D}_{\mathrm{u}}$ using Equation 9 and optimize the policy with any offline RL algorithm

---

model consists of three components: the encoder $q_\theta$, the decoder $p_\theta$, and a learnable prior $f_\theta$, which can be jointly optimized with the *Evidence Lower Bound (ELBO)*:

$$
\begin{aligned}
&\log p(\sigma_{t:t+k}|s_t, a_t) \\
&= \log \int q_\theta(z_t|s_t, a_t, \sigma_{t:t+k}) \frac{f_\theta(z_t|s_t, a_t) p_\theta(\sigma_{t:t+k}|s_t, a_t, z_t)}{q_\theta(z_t|s_t, a_t, \sigma_{t:t+k})} \mathrm{d}z_t \\
&\geq \mathbb{E}_{q_\theta(z_t|s_t, a_t, \sigma_{t:t+k})} \left[ \log \frac{f_\theta(z_t|s_t, a_t) p_\theta(\sigma_{t:t+k}|s_t, a_t, z_t)}{q_\theta(z_t|s_t, a_t, \sigma_{t:t+k})} \right] \\
&= \mathbb{E}_{q_\theta(z_t|s_t, a_t, \sigma_{t:t+k})} \left[ \log p_\theta(\sigma_{t:t+k}|s_t, a_t, z_t) \right] - \mathrm{KL} \left[ q_\theta(z_t|s_t, a_t, \sigma_{t:t+k}) \| f_\theta(z_t|s_t, a_t) \right] \\
&\stackrel{\text{def}}{=} -\mathcal{L}_{\mathrm{ELBO}}(s_t, a_t, \sigma_{t:t+k}; \theta),
\end{aligned} \tag{7}
$$

where the third line follows from Jensen's Inequality. Following the pre-training phase, the VAE can be utilized for both reward learning and reward labeling. During reward learning, the embedding $z_t$ can be employed as a substitute for $\sigma_{t:t+k}$ in the preference model:

$$
\rho_{\mathrm{HPM}}(\sigma; \psi) = \sum_{(s_t, a_t) \in \sigma} r_\psi(s_t, a_t | \sigma_{t:t+k}) \approx \sum_{(s_t, a_t) \in \sigma} r_\psi(s_t, a_t | z_t). \tag{8}
$$

Here, the embedding is obtained using the encoder $z_t \sim q_\theta(\cdot|s_t, a_t, \sigma_{t:t+k})$. Plugging this into the Bradley-Terry model gives an approximation of HPM (Equation 6). We then once again utilize the preference dataset along with the cross-entropy loss, as defined in Equation 2, to optimize $r_\psi$. During the reward labeling phase, we compute the reward using the prior distribution $f_\theta$:

$$
r_\psi(s_t, a_t) = \mathbb{E}_{z_t \sim f_\theta(\cdot|s_t, a_t)} \left[ r_\psi(s_t, a_t, z_t) \right] \approx \frac{1}{N} \sum_{l=1}^{N} r_\psi(s_t, a_t, z_t^{(l)}), \tag{9}
$$

where $z_t^{(1)}, z_t^{(2)}, \ldots, z_t^{(N)}$ are i.i.d. samples from $f_\theta$. With a large $N$, we can obtain approximations of the expected reward for downstream reinforcement learning.

We train these models using the unlabelled dataset $\mathcal{D}_{\mathrm{u}}$. This offers two benefits. Firstly, $\mathcal{D}_{\mathrm{u}}$ typically encompasses a substantial volume of data, which enhances model performance. Secondly, the scalar reward is obtained by marginalizing over the prior distribution $f_\theta$ during the reward labeling phase. Precisely aligning $f_\theta$ with $\mathcal{D}_{\mathrm{u}}$ can significantly enhance the stability of this marginalization process, particularly in instances of distributional shifts between $\mathcal{D}_{\mathrm{u}}$ and $\mathcal{D}_{\mathrm{p}}$.

**Practical Implementation.** We employ the GPT architecture (Brown et al., 2020) for the encoder $q_\theta$ due to its expressivity in sequence modeling. Given an input segment $\sigma = (s_0, a_0, s_1, a_1, \ldots, s_H, a_H)$, we concatenate $s_t$ and $a_t$ together as a single token. In each attention layer, we apply the anti-causal attention mask which restricts each token's attention to itself and

Table 1: Normalized averaged score for locomotion tasks (top) and manipulation tasks (bottom). In the table, "hop" is abbreviated for the Hopper task, "walk" for Walker2D, "ham" for Hammer, "m" for medium, "r" for replay, "e" for expert, "h" for human, "c" for cloned. The reference scores for MR and PT are from Kim et al. (2023), while those for IPL are from Hejna & Sadigh (2023). For the rest numbers, we use our own implementations and report the average and the standard deviation of the performances across 10 evaluation episodes and 5 seeds. We bolded values that are within 95% of the top-performing method among our implemented versions.

| Dataset | Oracle | SFT | MR | | PT | | IPL | | HPL (Ours) |
| --- | --- | --- | --- | --- | --- | --- | --- | --- | --- |
| | | | ref. | reimpl. | ref. | reimpl. | ref. | reimpl. | |
| hop-m-r | 97.4 | 22.2 | 11.6 | $64.3_{\pm 18.2}$ | 84.5 | $77.4_{\pm 8.0}$ | 73.6 | $56.1_{\pm 20.3}$ | $\mathbf{83.0_{\pm 14.4}}$ |
| hop-m-e | 107.4 | 5.2 | 57.8 | $86.3_{\pm 21.6}$ | 69.1 | $78.7_{\pm 27.8}$ | 74.5 | $67.8_{\pm 18.0}$ | $\mathbf{104.0_{\pm 7.7}}$ |
| walk-m-r | 82.2 | 9.0 | 72.1 | $\mathbf{69.5_{\pm 1.7}}$ | 71.3 | $64.0_{\pm 15.2}$ | 59.9 | $42.3_{\pm 17.4}$ | $64.1_{\pm 8.9}$ |
| walk-m-e | 111.7 | 0.4 | 108.3 | $90.8_{\pm 9.0}$ | 110.1 | $102.2_{\pm 17.5}$ | 108.5 | $\mathbf{106.1_{\pm 4.6}}$ | $\mathbf{108.9_{\pm 0.5}}$ |

| Dataset | Oracle | SFT | MR | PT | IPL | HPL (Ours) |
| --- | --- | --- | --- | --- | --- | --- |
| pen-h | 78.5 | 36.4 | $14.1_{\pm 9.0}$ | $11.2_{\pm 4.5}$ | $11.5_{\pm 11.6}$ | $\mathbf{70.9_{\pm 23.2}}$ |
| pen-c | 83.4 | 31.1 | $13.8_{\pm 4.8}$ | $11.9_{\pm 13.3}$ | $12.3_{\pm 6.6}$ | $\mathbf{33.1_{\pm 19.6}}$ |
| ham-h | 1.8 | 0.3 | $0.2_{\pm 0.0}$ | $0.2_{\pm 0.3}$ | $0.0_{\pm 0.0}$ | $\mathbf{4.3_{\pm 4.7}}$ |
| ham-c | 1.5 | **2.6** | $0.0_{\pm 0.1}$ | $2.0_{\pm 4.6}$ | $0.1_{\pm 0.1}$ | $0.3_{\pm 0.0}$ |
| drawer-open | - | 0.42 | $\mathbf{0.92_{\pm 0.10}}$ | $0.39_{\pm 0.24}$ | $0.54_{\pm 0.26}$ | $\mathbf{0.95_{\pm 0.07}}$ |
| button-press | - | 0.26 | $0.61_{\pm 0.04}$ | $0.38_{\pm 0.24}$ | $0.58_{\pm 0.11}$ | $\mathbf{0.70_{\pm 0.14}}$ |
| plate-slide | - | 0.26 | $0.38_{\pm 0.08}$ | $0.29_{\pm 0.27}$ | $0.34_{\pm 0.29}$ | $\mathbf{0.43_{\pm 0.13}}$ |
| sweep-into | - | 0.24 | $0.31_{\pm 0.10}$ | $0.22_{\pm 0.13}$ | $0.14_{\pm 0.15}$ | $\mathbf{0.37_{\pm 0.11}}$ |

subsequent tokens, ensuring that the output token $z_t$ encapsulates the forward-looking information starting from time step $t$. The decoder network $p_\theta$ reconstructs $\sigma_{t:t+k}$ using the embedding $z_t$ and $(s_t, a_t)$. In our implementation, $p_\theta$ takes inputs of $s_t, a_t, z_t$ and a time interval $\Delta t \in \{0, 1, ..., k\}$, and predicts $(s_{t+\Delta t}, a_{t+\Delta t})$. This parameterization facilitates the parallel decoding of the entire trajectory by processing all specified intervals in a single forward pass. Finally, the prior $f_\theta$ is parameterized as an MLP network which receives $(s_t, a_t)$ and outputs a distribution over the embedding space.

## 3.4 OVERALL FRAMEWORK OF HPL

Putting everything together, we outline *Hindsight Preference Learning (HPL)* in Algorithm 1. HPL can be divided into three stages: 1) pre-training a VAE to embed future segments, using data from the unlabeled dataset $\mathcal{D}_u$ (line 1-6); 2) training the conditional reward function $r_\psi$ with the preference dataset $\mathcal{D}_p$ (line 7-12); and finally 3) label the unlabeled dataset with Equation 9, followed by applying any offline RL algorithm for policy optimization (line 13-14).

## 4 RELATED WORK

**Preference-based Reinforcement Learning.** Human preferences are easier to obtain compared to well-calibrated step-wise rewards or expert demonstrations in some domains, making them a rich yet easy source of signals for policy optimization. Christiano et al. (2017) utilize the Bradley-Terry model to extract reward function from human preferences and lay the foundation for using deep RL to solve complex tasks. Based on this, several methods (Lee et al., 2021; Ibarz et al., 2018; Liang et al., 2022b) further improve the query efficiency by incorporating techniques like pre-training and relabeling. OPRL (Shin et al., 2023) further proposes principled rules for query selection and provides baseline results using existing offline datasets. On the other hand, some works bypass the need for a reward model. IPL (Hejna & Sadigh, 2023) achieves this by expressing the reward with value functions via the inverse Bellman operator, while OPPO (Kang et al., 2023) uses Hindsight Information Matching (HIM) to conduct preference learning in compact latent space. FTB (Zhang et al., 2023) employs powerful generative models to diffuse bad trajectories to preferred ones. DPPO (An et al., 2023) and CPL (Hejna et al., 2023), although with different starting points, both directly optimize the policy by relating it to the preferences.

**Human Preference Modeling.** To extract utilities from human preferences for policy optimization, we need preference models to establish the connection between preferences and utilities. A common approach is to use the Bradley-Terry model (Christiano et al., 2017) and hypothesizes that preference is emitted according to the sum of Markovian rewards, while Preference Transformer (Kim et al., 2023) and Hindsight PRIOR (Verma & Metcalf, 2024) extend this by using the weighted sum of non-Markovian rewards. Besides, another line of research proposes that human preference is decided by the sum of optimal advantages in the segment (Knox et al., 2022; 2024) rather than the rewards. In this paper, we focus on the influence of the future and consider the sum of future-conditioned rewards.

**Leveraging Hindsight Information.** Hindsight information can provide extra supervision during training. For example, HER (Andrychowicz et al., 2017) and its follow-up works (Eysenbach et al., 2020; Li et al., 2020; Zhang & Stadie, 2022) relabel the transitions to allow sample-efficient learning in sparse-reward or goal-reaching tasks. Prior works have also explored learning representations by predicting the future (Furuta et al., 2022; Xie et al., 2023; Yang et al., 2023), and such representations facilitate downstream tasks such as policy optimization (Furuta et al., 2022; Xie et al., 2023), preference modeling (Kang et al., 2023) and exploration (Jarrett et al., 2023). Perhaps the most related works are HCA (Harutyunyan et al., 2019) and CCA (Mesnard et al., 2021), which all propose to make the value functions dependent on the future and derive corresponding policy gradients with a lower variance. The focus of HCA and CCA lies in temporal credit assignment, while HPL elucidates the impact of future information within the context of PbRL and explores ways to elicit better rewards from preference comparisons.

## 5 EXPERIMENTS

We evaluate HPL as well as other methods with various benchmarks. Specifically, we selected two tasks (Hopper and Walker2D) from Gym-MuJoCo locomotion (Brockman et al., 2016), two tasks (Hammer and Pen) from the Adroit manipulation platform (Kumar, 2016), and four tasks (Drawer-Open, Button-Press, Plate-Slide and Sweep-Into) from Meta-World Benchmark (Yu et al., 2020). For Gym-MuJoCo tasks and Adroit tasks, we select datasets from the D4RL Benchmark (Fu et al., 2020) and mask the reward labels as the unlabeled dataset $\mathcal{D}_u$, while the annotated preference dataset $\mathcal{D}_p$ is provided by Kim et al. (2023). For Meta-World tasks, we used the datasets released by Hejna & Sadigh (2023) as $\mathcal{D}_u$ and $\mathcal{D}_p$. It is worthwhile

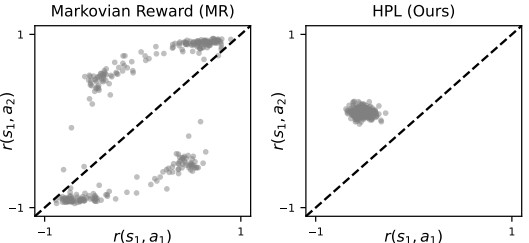

Figure 3: The rewards values given by the MR method and HPL. Each dot represents one trial and its coordinates are the estimated rewards.

to note that for Gym-MuJoCo and Adroit tasks, the preference label is generated by real human annotators, while for Meta-World tasks it is synthesized based on trajectory return. More details about the datasets and how the preference annotations are generated can be found in Appendix A.

For baseline methods, we consider popular algorithms such as 1) **Oracle**, which uses the oracle step-wise reward for policy optimization; 2) Supervised Fine-Tuning (**SFT**), which imitates the preferred segments; 3) **MR**, which uses the Bradley-Terry Model to extract Markovian rewards from the preferences; 4) Preference Transformer (**PT**) (Kim et al., 2023), which uses a transformer and bidirectional layer to model the reward; and 5) Inverse Preference Learning (**IPL**) (Hejna & Sadigh, 2023), which removes the need of reward modeling using inverse Bellman operator. Besides, we also compare HPL with two other direct alignment methods, **CPL** and **OPPO** in Appendix E.1. More details about the implementations can be found in Appendix B. For evaluation metrics, we report the normalized score for MuJoCo and Adroit tasks and the success rate for Meta-World tasks.

### 5.1 BENCHMARK RESULTS

Our first experiment investigates the capability of HPL in standard offline PbRL settings using both $\mathcal{D}_u$ and $\mathcal{D}_p$. For policy optimization, we used IQL for all methods except for SFT. We found certain

design choices such as reward normalization can have a significant effect on the performance, so we included the reference score (denoted as ref.) from the original paper for some algorithms and the score of our implementation (denoted as reimpl.) for fair comparisons.

The results are listed in Table 1. We also implement variants that use AWAC for policy optimization, and the results are deferred to Appendix E.2. Overall, HPL consistently outperforms other baselines both in locomotion tasks and manipulation tasks, especially in complex domains like the pen task. The promising performance validates the effectiveness of HPL for learning from human preferences.

## 5.2 Tasks with Preference Distribution Shift

As we illustrated in Section 3.1, the distribution mismatch between the preference dataset $\mathcal{D}_\mathrm{p}$ and the unlabeled dataset $\mathcal{D}_\mathrm{u}$ may affect credit assignments. We take the gambling MDP (Figure 2) as a sanity check to see whether HPL can deliver better results. We used the dataset $\mathcal{D}$ in Section 3.1 as the preference dataset $\mathcal{D}_\mathrm{p}$, and additionally collected $\mathcal{D}_\mathrm{u}$ by randomly choosing between $a_1$ and $a_2$. Afterwards, we compare the rewards of $(s_1, a_1)$ and $(s_1, a_2)$ given by both MR and HPL. We run both algorithms for 500 random seeds and plot the results in Figure 3. In the figure, each point stands for one trial and its coordinate stands for $r_\psi(s_1, a_1)$ and $r_\psi(s_1, a_2)$ respectively. We note that every trial of both methods has achieved $100\%$ accuracy for predicting the preference labels, so we focus on the quality of rewards. As discussed in Section 3.1, successful credit assignment should try to assign lower values to $r(s_1, a_1)$, i.e. the point should lie in the above triangular area. While HPL steadily assigns higher rewards to $(s_1, a_2)$, MR over-estimates the rewards for $(s_1, a_1)$ in almost half of the cases.

We explain the success of HPL by taking a closer look at how HPL adjusts and achieves credit assignment. By incorporating the future state ($s_\mathrm{good}$ or $s_\mathrm{bad}$), we see in Figure 4a that HPL effectively identifies two different outcomes at $(s_1, a_1)$ and assign different values for $r(s_1, a_1|s_\mathrm{good})$ and $r(s_1, a_1|s_\mathrm{bad})$. Figure 4b plots the transition probabilities $\hat{p}(s_\mathrm{good}|s_1, a_1)$ and $\hat{p}(s_\mathrm{good}|s_1, a_1)$ estimated by the prior network $f_\theta$. By sampling $z$ according to the probabilities, we observe in Figure 4c that at $(s_1, a_1)$, the negative outcome is emphasized since the agent transits to $s_\mathrm{bad}$ for most of the time. Ultimately, HPL achieves the correct credit assignment, i.e. $r(s_1, a_1) < r(s_1, a_2)$. Although it is still possible that MR can find the optimal policy through RL optimization, we emphasize that HPL delivers more robust and advantageous rewards that will facilitate subsequent policy optimization.

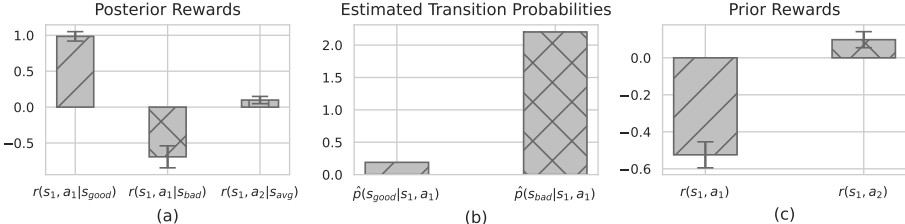

Figure 4: Internal reward adjustment process of HPL. HPL adjusts the posterior reward (a) using the transition probabilities by the prior network (b) and yields better credit assignment (c).

**Larger Scale Experiments with Preference Distribution Shift.** To further validate HPL's capability of learning better rewards in the face of the preference distribution shift on a larger scale, we constructed a series of tasks by combining $\mathcal{D}_\mathrm{u}$ and $\mathcal{D}_\mathrm{p}$ collected by different behavior policies. For example, "hopper: med-e → med" means we used $\mathcal{D}_\mathrm{p}$ with *medium-expert* quality and $\mathcal{D}_\mathrm{u}$ with *medium* quality. The performance curves are presented in Figure 5. In such mismatched scenarios, HPL performs better than all baseline algorithms in most of the tasks. Besides, HPL demonstrates faster convergence speeds and more stable performances, which validates that the reward from HPL is more robust and shaped for downstream RL optimization.

## 5.3 Analysis of HPL

In this section, we examine each part of HPL to gain a deeper understanding of each design choice.

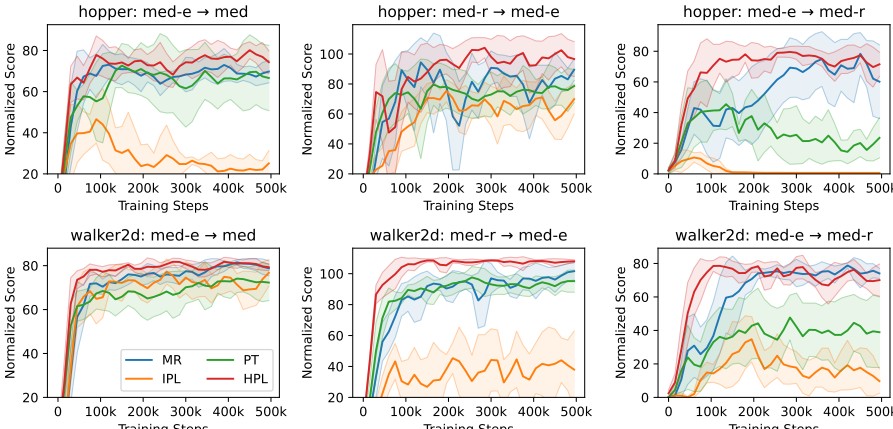

Figure 5: The performance curves of HPL and baseline methods in tasks with mismatched datasets. We report the average (solid line) and the standard deviation (shaded area) of each algorithm across 5 random seeds and 10 evaluation episodes for each seed.

**Future Segment Embedding and the VAE Structure.** HPL relies on the VAE structure to generate compact embeddings for future segment representation and sampling. Consequently, our first analysis investigates the quality of these embeddings. Figure 6 displays the images of a trajectory segment from the offline dataset (top left) and its reconstruction by the VAE (bottom left). Note that both the encoding and reconstruction processes are based on states and actions, rather than pixel observations. The VAE reconstruction is highly accurate, indicating that the embedding $z_t$ effectively compresses the relevant information. In the right figure, we select one $(s_t, a_t)$ from the offline dataset, sample embeddings $z_t$ from the prior $f_\theta$, and decode the trajectories $\sigma_{t:t+k}$. We compute the embedding log-probability $\log p(z_t|s_t, a_t) = \log f_\theta(z_t|s_t, a_t))$ and also the trajectory log-probability $\log p(\sigma_{t:t+k}|s_t, a_t) = \sum_{i=1}^{k} \log \pi_\beta(a_{t+i}|s_{t+i})$ with additionally trained behavior cloning policy $\pi_\beta$. The positive correlation observed in Figure 6 between these two log probabilities validates the efficacy of sampling from $f_\theta$.

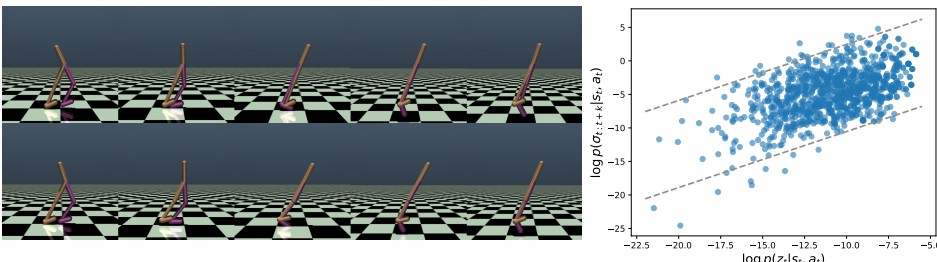

Figure 6: Left: The rendered image of one raw trajectory selected from the offline dataset (top row) and the reconstruction by the VAE (bottom row). Right: The relationship between the log-probabilities of segments and their embeddings.

**Ablation on the Future Length $k$.** The parameter $k$ controls the length of future segments encoded into the embedding $z$. As illustrated in Figure 7a (results of all four tasks in Figure 11), extending $k$ generally enhances performance, highlighting the benefits of future conditioning. However, as $k$ exceeds a certain threshold, we witness fluctuations and decreases in the performances, probably due to the challenges of representing longer future segments accurately.

**Scaling with Dataset Sizes.** We conduct experiments to evaluate the scalability of MR and HPL with varying dataset sizes. In Figure 7b (results of all four tasks in Figure 13), we adjust the size of $\mathcal{D}_u$ from 10% to 100% of its total capacity and observe that HPL consistently outperforms MR across all proportions of unlabeled data. Figure 7c (results of all four tasks in Figure 12) illustrates the scaling trends of HPL and MR with different numbers of preference queries. While both methods exhibit an upward trend in success rates, HPL is comparatively better under various data scales.

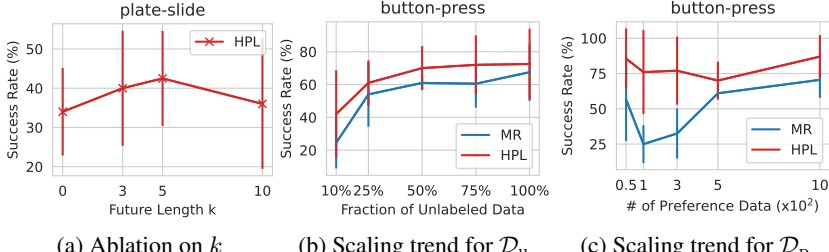

Figure 7: Quantitative analysis of HPL. We report the mean (solid line) and the standard deviation (error bar) across 10 seeds for all experiments.

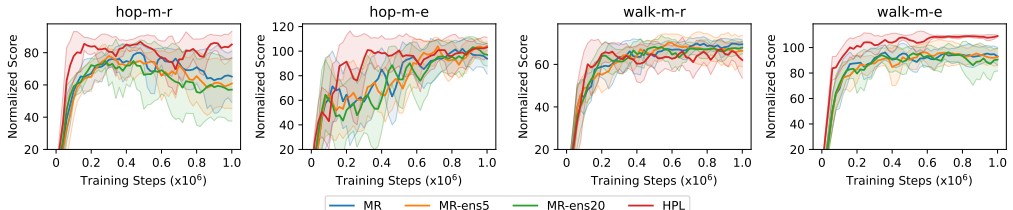

Figure 8: Learning curve of HPL and MR variants with ensemble reward networks. We report the average and the standard deviation of the performances across 10 evaluation episodes and 5 seeds.

**Ablation on the Ensemble Effect.** One may argue that the success of HPL comes from the marginalization step (Equation 9), which implicitly ensembles reward models to yield improved rewards for subsequent RL optimization. Indeed, the reward model ensemble does bring benefit to the credit assignment by characterizing the aleatoric uncertainty and thus facilitating active knowledge acquisition (Liang et al., 2022a) or promoting pessimism (Coste et al., 2024). To ablate this effect, we apply the ensemble trick to the MR method, by using an ensemble of 5 and 20 reward models while keeping other configurations unchanged. These reward models are trained with the same dataset, differing only in their initialization. When labeling the dataset, we take the average of the outputs of the ensembles as the reward. Note that this can be considered as an implementation of OPRL (Shin et al., 2023), which also employs the ensemble technique.

As witnessed in Figure 8, MR still falls behind HPL despite the ensemble technique. This justifies that, naively ensembling reward models which are trained via different initialization does not benefit the credit assignment. On the other hand, HPL ensembles reward values conditioned on different future outcomes based on their prior probabilities, producing more reliable and advantageous rewards.

## 6   CONCLUSIONS AND DISCUSSIONS

This paper focuses on extracting rewards from human preferences for RL optimization. Unlike previous methods that assume the preference is determined by the sum of Markovian rewards, our method, HPL, instead employs a new preference model that correlates the preference strength with the sum of rewards which are conditioned on the future outcome of this trajectory. By marginalizing the conditional reward over the prior distribution of future outcomes induced by the vast unlabeled dataset, HPL produces more robust and suitable reward signals for downstream RL optimization.

**Limitations.** The primary limitation of the current version of HPL lies in its failure to exploit the full potential of the learned VAE. As a generative model, VAE could possibly be employed to augment the training data via sampling, estimate reward uncertainty (Liang et al., 2022a) using the learned prior distribution, or identify diverse preferences (Xue et al., 2023) automatically from the data. Besides, HPL implicitly assumes the reward and the distribution of future segments possess the MDP property, i.e. they are determined by the immediate step $(s_t, a_t)$. Inspired by the design of world models such as Dreamer-v2 (Hafner et al., 2021) and PlaNet (Hafner et al., 2019), we can model the dependence of history by using RNNs to encode the history into embeddings and feed the embeddings to the VAE structures. Future work will focus on extending the HPL framework to broader applications.

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

# A TASKS AND DATASETS

## A.1 TASKS

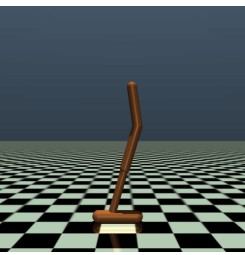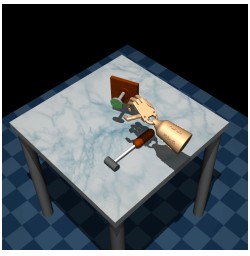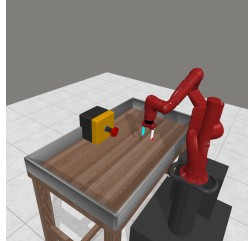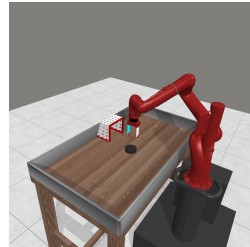

Figure 9: From left to right, the figures show the Hopper task from Gym-MuJoCo, the Hammer task from the Adroit platform, and the button-press and plate-slide tasks from Meta-World.

We evaluated the HPL algorithm on different environments, including Gym-MuJoCo (Brockman et al., 2016), Adroit (Kumar, 2016), and Meta-World (Yu et al., 2020). Figure 9 provides illustrations for the environments used in our evaluation. These tasks range from basic locomotion to complex manipulation. Among them, Gym-MuJoCo and Meta-World are released with an MIT license, while Adroit is released with the Apache-2.0 license.

**Gym-MuJoCo.** We selected the Hopper and Walker2D tasks from the Gym-MuJoCo environment. The goal of the Hopper task is to control a single-legged robot to hop forward, with primary rewards based on forward speed, energy consumption, and a survival bonus for maintaining stability. The Walker2D task involves controlling a bipedal robot to walk forward, while the rewards are designed based on the forward speed, control penalties, and a survival bonus. The key challenge in both tasks is to maximize the forward distance while maintaining the robot's stability.

**Adroit.** We chose the Hammer and Pen tasks from the Adroit environment. These tasks require controlling a 24-DoF simulated Shadow Hand robot to perform precise manipulations. The Hammer task involves using the robot to hammer a nail, with rewards given for successful strikes and penalties for misses or ineffective actions. The Pen task requires the robot to rotate a pen, rewarding successful rotations and penalizing failures or instability. Adroit tasks emphasize high precision and the complexity of robotic hand manipulations.

**Meta-World.** We selected multiple manipulation tasks from the Meta-World environment, including drawer-open, button-press, plate-slide, and sweep-into. These tasks require a Sawyer robotic arm to perform multi-step operations. For example, the drawer-open task involves grasping and pulling open a drawer, the button-press task requires accurately pressing a designated button, the plate-slide task involves pushing a plate to a specified location, and the sweep-into task requires sweeping objects into a target area. The reward structure in these tasks is designed as a combination of sub-tasks, providing partial rewards for each sub-task completed and a total reward for successfully completing the entire task. Meta-World tasks highlight the shared structure between tasks and the sequential nature of complex operations.

## A.2 UNLABELED OFFLINE DATASETS

For the unlabeled offline dataset $\mathcal{D}_{\mathrm{u}}$, we used the datasets provided in D4RL (Fu et al., 2020) for Gym-MuJoCo and Adroit tasks and the datasets from Hejna & Sadigh (2023) for Meta-World tasks.

**Gym-MuJoCo Datasets.** The datasets for Hopper and Walker2D tasks were obtained through online training and include *medium*, *medium-replay*, and *medium-expert* datasets. The *medium* dataset was generated by training a policy using Soft Actor-Critic (Haarnoja et al., 2018), stopping early when the policy reached a medium performance level, and collecting 1 million samples from this partially trained policy. The *medium-replay* dataset contains all samples observed in the replay buffer during training until the policy reaches medium performance. The *medium-expert* dataset was created by mixing equal amounts of expert demonstration data and medium-level policy data. All of these

datasets can be obtained following the APIs provided by D4RL, and the datasets are licensed with the CC BY 4.0 license.

**Adroit Datasets.** We used data for the Hammer and Pen tasks. These datasets include *human*, *expert*, and *cloned* datasets. The *human* dataset consists of a small number of demonstrations collected from human experts, with each task containing 25 trajectories. The *expert* dataset comprises a large amount of expert data collected from fine-tuned RL policies. The *cloned* dataset is generated by training an imitation policy on the demonstration data, running this policy, and mixing the generated data with the original demonstrations in a 50-50 ratio. This data generation method simulates a real-world scenario where a small amount of human demonstration data is augmented through imitation learning. All of these datasets can be obtained following the APIs provided by D4RL, and the datasets are licensed with the CC BY 4.0 license.

**Meta-World Datasets.** We used the preference-annotated dataset from Hejna & Sadigh (2023) and converted it into an unlabeled offline dataset by discarding the reference label. These datasets can be obtained following the original source of Hejna & Sadigh (2023). The authors did not specify the license of the datasets. However, the codes are released with the MIT license so we speculate the datasets inherit the MIT license as well since they are released together.

### A.3 PREFERENCE DATASETS

For the preference dataset $\mathcal{D}_\mathrm{p}$, we selected the human-annotated datasets from Kim et al. (2023) for Gym-MuJoCo and Adroit tasks, and the synthetic datasets from Hejna & Sadigh (2023) for Meta-World tasks. The datasets are released alongside with the codes (`https://github.com/csmile-1006/PreferenceTransformer` and `https://github.com/jhejna/inverse-preference-learning` respectively). The authors did not specify the license of the datasets. However, the codes are both released with the MIT license so we speculate the datasets inherit the MIT license as well since they are released together. In the following paragraphs, we detail the construction of these preference datasets based on the details provided by their original creators.

For Gym-MuJoCo datasets, preferences were collected from actual human subjects. Specifically, human annotators watched the rendered videos of segments and selected the segment they believed was more helpful in achieving the agent's goal. Each segment lasted 3 seconds (100 frames). Human annotators can prefer one of the segment pairs or remain neutral by assigning equal preference to both segments. The annotators are instructed to make decisions based on some criteria. For the Hopper task, the robot is expected to move to the right as far as possible while minimizing energy consumption. Segments, where the robot lands unstably, are rated lower, even if the distance traveled is longer. If two segments are nearly tied on this metric, the one with the greater distance is chosen. For the Walker2D task, the goal is to move the bipedal robot to the right as far as possible while minimizing energy consumption. Segments where the robot is about to fall or walks abnormally (e.g., using only one leg or slipping) are rated lower, even if the distance covered is longer. If two segments are nearly tied on this metric, the one with the greater distance is chosen. For the *medium-replay* offline dataset, there are 500 queries, while for the *medium-expert* offline dataset, there are 100 queries in total. The segment length for all datasets is $H = 100$.

The Meta-World datasets included script preferences that came from Hejna & Sadigh (2023). First, the datasets included 100 trajectories of expert data for the target task, adding Gaussian noise with a standard deviation of 1.0. Then, the datasets included 100 trajectories of sub-optimal data by running the ground truth policy with noise on different randomizations of the target task, and another 100 trajectories of sub-optimal data by running the ground truth policy of different tasks within the target domain with noise. Finally, the datasets included 100 trajectories generated using uniform random actions. Each Meta-World task dataset contains 200k time steps. The preference datasets were constructed by uniformly sampling segments and assigning preference labels based on the total rewards of the segments.

## B ALGORITHM IMPLEMENTATIONS

In this section, we detail the implementations of both HPL and the baseline algorithms used in this paper.

## B.1 PREFERENCE LEARNING METHODS

**Markovian Reward (MR).** The MR method optimizes a Markovian reward function $r_\psi(s, a)$ using the Bradley-Terry model and the preference dataset $\mathcal{D}_p$. The hyper-parameters for MR are listed in Table 2. It is worth noting that we add a final activation layer to the reward network to scale the reward to $[0, 1]$. We find that without such activation, the performance of RL severely deteriorates in some of the Gym MuJoCo tasks. We suspect that this is related to the *survival instinct* (Li et al., 2023) in offline RL, i.e. in environments with terminal conditions, negative rewards tend to incline the agent to terminate the trajectory by selecting those dangerous out-of-distribution actions. Based on this observation, we decided to activate the rewards with Sigmoid for the Hopper and Walker2D tasks while leaving them unchanged for other tasks without environmental terminations. Such activation is shared across MR, PT, and HPL. However, one may argue that the activation implicitly imposes an inductive bias on the obtained reward, which may not align with the ground truth. So we also add the reference scores in Section 5.1 for Gym MoJoCo tasks for comprehensive comparisons.

Table 2: Hyper-parameters for MR.

| | |
|---|---|
| hidden dimension for $r_\psi$ | 256 |
| # of hidden layers for $r_\psi$ | 2 for Gym MuJoCo tasks and 3 for others |
| final activation | Sigmoid for Gym MuJoCo tasks and Identity for others |
| optimizer | Adam |
| learning rate | 0.0003 |
| training steps for $r_\psi$ | 50k |

**Preference Transformer (PT).** PT employs a causal transformer followed by a bi-directional attention layer to estimate the rewards. Using the causal transformer, the states and actions can attend to historical tokens and thus the reward can utilize the historical information. The final bi-directional attention layer uses the attention scores as the weights of the rewards at each time step. The authors found PT can identify and place more emphasis on those critical states and actions. We also re-implemented the PT based on the original Jax implementation provided by the authors, and the hyper-parameters are listed in Table 3. Note that we do not use any validation to select the reward model.

Table 3: Hyper-parameters for PT.

| | |
|---|---|
| attention embedding dimension | 256 |
| # of attention layers | 3 |
| # of attention heads | 1 |
| dropout rate | 0.1 |
| final activation | Sigmoid for Gym MuJoCo tasks and Identity for others |
| optimizer | Adam |
| learning rate | 0.0003 |
| learning rate warm-up steps | 10k |
| training steps for $r_\psi$ | 100k |

**Inverse Preference Learning (IPL).** IPL removes the need for learning a reward model, by expressing the reward using the value functions $Q(s, a)$ and $V(s)$ of the RL agent:

$$r(s, a) = Q(s, a) - \gamma \mathbb{E}_{s' \sim T(s'|s,a)} \left[ V(s') \right]. \tag{10}$$

By substituting Equation 10 into Equation 4, the loss $\mathcal{L}_{\text{MR}}$ provides guidance to increase the Q-values of preferred states and actions. We also re-implement IPL in this paper and keep the hyper-parameters of IPL the same as the ones used in the original paper.

**Hindsight Preference Learning (HPL).** The key components of HPL are the conditional reward model $r_\psi$ and the VAE. We list the hyper-parameters of these modules in Table 4. The hyper-parameters are kept the same as listed in Table 4 unless otherwise noted.

Table 4: Hyper-parameters for HPL.

| | | |
|---|---|---|
| VAE | attention embedding dim of encoder $q_\theta$ | 256 |
| | # of attention layers | 3 |
| | # of attention heads | 1 |
| | dropout rate | 0.1 |
| | hidden dim for decoder $p_\theta$ | 256 for MuJoCo/Adroit, 512 for Meta-World |
| | # of hidden layers for decoder $p_\theta$ | 256 |
| | hidden dim for prior $f_\theta$ | 256 for MuJoCo/Adroit, 512 for Meta-World |
| | # of hidden layers for prior $f_\theta$ | 2 |
| | dimension of embedding $z$ | 128 |
| | posterior/prior distribution | categorical distribution |
| | learning rate | 3e-4 |
| | training steps | 250k |
| | KL loss coefficient | 0.1 |
| | encoded future segment length $k$ | 5 |
| $r_\psi$ | hidden dims of $r_\psi$ | 256 |
| | # of hidden layers for $r_\psi$ | 3 |
| | final activation | Sigmoid for MuJoCo, Identity for others |
| | optimizer | Adam |
| | learning rate | 3e-4 |
| | training steps | 100k |
| | marginalization samples $N$ | 20 |

## B.2 RL POLICY OPTIMIZATION

For those methods that follow the two-phase paradigm as we discussed in Section 2.2, we use Implicit Q-Learning (IQL) (Kostrikov et al., 2022) for policy optimization with the learned reward model. The hyper-parameters for IQL are shared across various reward learning methods for fair comparisons. We list the hyper-parameters in Table 5.

Table 5: Hyper-parameters for IQL.

| | |
|---|---|
| expectile $\tau$ | 0.7 for MuJoCo, 0.75 for others |
| inverse temperature | 0.333 |
| clipping threshold for $\exp A$ | 100 |
| discounting factor | 0.99 |
| soft update for target networks | 0.005 |
| policy network | $\texttt{MLP}(\dim(\mathcal{S}), 256, 256, 256, 2*\dim(\mathcal{A}))$ |
| policy distribution | tanh-squashed diagonal Gaussian |
| critic network | $\texttt{MLP}(\dim(\mathcal{S}) + \dim(\mathcal{A}), 256, 256, 256, 1)$ |
| value network | $\texttt{MLP}(\dim(\mathcal{S}), 256, 256, 256, 1)$ |
| optimizers for all networks | Adam |
| learning rates for all networks | 0.0003 |
| training steps | 500k |

## B.3 SUPERVISED FINE-TUNING

Finally, we provide details about the implementations of the Supervised Fine-Tuning (SFT) method used in the experiment section. For SFT, we use the preferred trajectory segments to perform behavior cloning. The behavior cloning process maximizes the log probability of the policy selecting the preferred segment. Thus, SFT methods fail to leverage the vast offline datasets, which is identified as a key advantage of offline PbRL methods. The hyper-parameters of SFT can be found in Table 6.

Table 6: Hyper-parameters for SFT.

| policy network | $\mathtt{MLP}(\dim(\mathcal{S}), 256, 256, 256, \dim(\mathcal{A}))$ |
|---|---|
| policy distribution | deterministic (Dirac distribution) |
| optimizer | Adam |
| learning rate | 0.0003 |
| training steps | 500k |

## C EXPERIMENTAL SETUPS

In this section, we provide additional details for the main results in Section 5.

**Benchmark results (Table 1).** We use the full amount of preference datasets and unlabeled datasets as detailed in Section A for MuJoCo tasks and Adroit tasks. For Meta-World tasks, we take the first 500 queries as the preference dataset $\mathcal{D}_\mathrm{p}$ and the first 5000 queries as the unlabeled dataset $\mathcal{D}_\mathrm{u}$. The $\mathcal{D}_\mathrm{p}$ and $\mathcal{D}_\mathrm{u}$ match each other in terms of the data source.

**Results of the mismatched tasks (Figure 5).** We created a series of tasks by cross-matching the preference datasets and the unlabeled datasets, as detailed in the main text. Besides, we use the full amount of the datasets, without further selection.

**Scaling trends with varied sizes of $\mathcal{D}_\mathrm{u}$ (Figure 7b).** In this set of experiments, we keep the setups and the hyper-parameters exactly the same as in Table 1, except for the sizes of the unlabeled dataset. Specifically, we select the first 1k, 2.5k, 5k, 7.5k, and 10k trajectories from the dataset (which correspond to 10%, 25%, 50%, 75%, and 100% of the total capacity of $\mathcal{D}_\mathrm{u}$).

**Scaling trends with varied sizes of $\mathcal{D}_\mathrm{p}$ (Figure 7c).** In this set of experiments, we keep the setups and the hyper-parameters exactly the same as in Table 1, except for the sizes of the preference dataset. We select the first 100, 200, 300, 400, 500, and 1000 queries from the dataset, as depicted in the figure.

## D DISCLOSURE OF COMPUTATIONAL RESOURCES AND EFFICIENCY

Throughout the experiments, we evaluate HPL as well as other baseline methods with workstations equipped with NVIDIA RTX 4090 cards. The running time of each method for the button-press task in the Meta-World environment is presented in Figure 10.

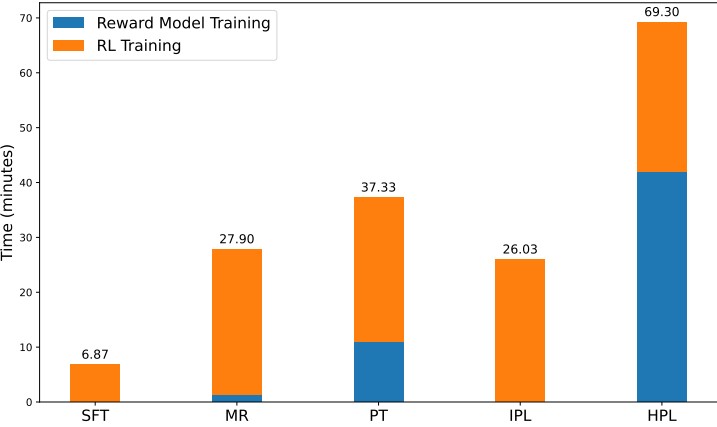

Figure 10: The running time of HPL and baseline methods, using button-press from Meta-World as an example.

# E  SUPPLIMENTARY EXPERIMENT RESULTS

Due to the limited space of the main text, we present additional supplementary results in this section.

## E.1  COMPARISON BETWEEN HPL AND ADDITIONAL BASELINES

Table 7: Normalized averaged score for HPL, SFT, OPPO, and CPL. In the table, we use the same abbreviations for tasks as in Table 1. We report the average and the standard deviation of the performances across 10 evaluation episodes and 5 seeds, and bold the values that are within 95% of the top-performing methods.

| Dataset | SFT | OPPO | CPL | HPL |
|---------|-----|------|-----|-----|
| hop-m-r | 22.2 | $37.4_{\pm 20.7}$ | $25.0_{\pm 9.6}$ | $\mathbf{83.3_{\pm 14.4}}$ |
| hop-m-e | 5.2 | $16.4_{\pm 6.5}$ | $54.7_{\pm 1.9}$ | $\mathbf{104.0_{\pm 7.7}}$ |
| walk-m-r | 9.0 | $44.9_{\pm 15.5}$ | $16.6_{\pm 5.5}$ | $\mathbf{64.1_{\pm 8.9}}$ |
| walk-m-e | 0.4 | $78.1_{\pm 20.2}$ | $78.5_{\pm 5.8}$ | $\mathbf{108.9_{\pm 0.5}}$ |
| drawer-open | 0.42 | - | $0.48_{\pm 0.2}$ | $\mathbf{0.95_{\pm 0.07}}$ |
| button-press | 0.26 | - | $0.05_{\pm 0.05}$ | $\mathbf{0.70_{\pm 0.14}}$ |
| plate-slide | 0.26 | - | $\mathbf{0.41_{\pm 0.23}}$ | $\mathbf{0.43_{\pm 0.13}}$ |
| sweep-into | 0.24 | - | $0.03_{\pm 0.04}$ | $\mathbf{0.37_{\pm 0.11}}$ |

In the main text, we compare HPL to popular reward-model-based algorithms such as PT. Recently, a wide range of direct alignment methods (Hejna et al., 2023; Kang et al., 2023; Rafailov et al., 2023; An et al., 2023) have been proposed to circumvent the onerous RL process and directly optimize the policies to align with human preferences. Among them, we focus on Contrastive Preference Learning (**CPL**) (Hejna et al., 2023), which is conceptually similar to both DPO (Rafailov et al., 2023) and DPPO (An et al., 2023); and also **OPPO** (Kang et al., 2023), which establishes a connection between high-dimensional trajectories and compact latent embeddings and optimizes the preferences in the latent space.

For the OPPO method, we employed the open-source code provided by the authors without any alterations, utilizing the same hyperparameters outlined in the original paper. For CPL, due to the limited size of our preference dataset (<500 pairs as compared to 20k pairs in CPL paper), vanilla CPL will lead to degraded performance. To mitigate this issue, we implement BC-Regularized CPL, which incorporates an auxiliary objective that behavior-clones the trajectories from the vast offline dataset to prevent performance collapse.

A summary of the performances is provided in Table 7. Although OPPO and CPL both outperform SFT, they still lag behind HPL by a large margin. Additionally, we observe that OPPO exhibits significant fluctuations in its learning curve and a decline in performance over time. Given these findings, we contend that in scenarios characterized by limited preference data, reinforcement learning algorithms may offer advantages over direct methods, as they provide enhanced generalization capabilities.

## E.2  BENCHMARK RESULTS OF AWAC VARIANTS

In Section 5.1, the results are obtained by using IQL (Kostrikov et al., 2022) as the policy optimization algorithm. However, given that IQL relies on expectile regression rather than the policy for bootstrapping, it may not fully reveal the potential shortcomings of the learned rewards. Additionally, the choice of expectile could significantly affect the outcomes. In this section, we instead implement AWAC, another offline reinforcement learning algorithm that integrates policy into bootstrapping, to both HPL and the baseline algorithms. This approach aims to provide a more thorough assessment of reward quality.

The results are listed in Table 8. Similar to HPL-IQL, HPL-AWAC demonstrates stable and consistent advantages over the baseline methods in most of the tasks. Overall, HPL-AWAC achieves the best performance on average across these three task suites.

Table 8: Normalized averaged score for AWAC variants of HPL and baseline algorithms. In the table, we use the same abbreviations for tasks as in Table 1. We report the average and the standard deviation of the performances across 10 evaluation episodes and 5 seeds, and bold the values that are within 95% of the top-performing methods among all methods except for the *Oracle*.

| Dataset | Oracle | SFT | MR-AWAC | PT-AWAC | IPL-AWAC | HPL-AWAC |
|---|---|---|---|---|---|---|
| hop-m-r | 97.4 | 22.2 | $31.2_{\pm0.2}$ | $68.7_{\pm18.3}$ | $69.8_{\pm13.6}$ | $\mathbf{94.6_{\pm3.1}}$ |
| hop-m-e | 107.4 | 5.2 | $70.9_{\pm34.5}$ | $\mathbf{93.3_{\pm13.9}}$ | $55.3_{\pm17.2}$ | $\mathbf{98.0_{\pm15.9}}$ |
| walk-m-r | 82.2 | 9.0 | $63.1_{\pm9.1}$ | $\mathbf{77.6_{\pm5.4}}$ | $8.9_{\pm11.0}$ | $71.2_{\pm4.0}$ |
| walk-m-e | 111.7 | 0.4 | $\mathbf{91.7_{\pm38.7}}$ | $76.7_{\pm47.4}$ | $46.3_{\pm52.8}$ | $\mathbf{95.1_{\pm8.5}}$ |
| Gym average | 99.7 | 9.2 | 64.2 | 79.1 | 45.1 | **89.7** |
| pen-h | 78.5 | 35.4 | $10.5_{\pm9.9}$ | $0.0_{\pm3.9}$ | $11.7_{\pm8.2}$ | $\mathbf{44.7_{\pm27.6}}$ |
| pen-c | 83.4 | 31.1 | $8.9_{\pm11.8}$ | $12.9_{\pm14.3}$ | $13.0_{\pm17.6}$ | $\mathbf{36.4_{\pm27.8}}$ |
| ham-h | 1.8 | 0.3 | $0.3_{\pm0.5}$ | $0.0_{\pm0.1}$ | $0.0_{\pm0.2}$ | $\mathbf{4.7_{\pm5.9}}$ |
| ham-c | 1.5 | **2.6** | $0.1_{\pm0.2}$ | $0.1_{\pm0.1}$ | $0.1_{\pm0.1}$ | $0.2_{\pm0.0}$ |
| Adroit average | 41.3 | 17.4 | 5.0 | 3.3 | 6.2 | **21.5** |
| drawer-open | - | 0.42 | $0.77_{\pm0.28}$ | $0.56_{\pm0.29}$ | $0.58_{\pm0.19}$ | $\mathbf{0.89_{\pm0.07}}$ |
| button-press | - | 0.26 | $\mathbf{0.78_{\pm0.14}}$ | $0.67_{\pm0.23}$ | $0.66_{\pm0.26}$ | $0.69_{\pm0.12}$ |
| plate-slide | - | 0.26 | $0.35_{\pm0.25}$ | $0.07_{\pm0.10}$ | $\mathbf{0.52_{\pm0.18}}$ | $0.47_{\pm0.21}$ |
| sweep-into | - | 0.24 | $0.30_{\pm0.19}$ | $0.10_{\pm0.12}$ | $0.24_{\pm0.11}$ | $\mathbf{0.49_{\pm0.09}}$ |
| Meta-World average | - | 0.30 | 0.55 | 0.35 | 0.50 | **0.64** |

### E.3 Ablation on Future Length $k$

The parameter $k$ controls the lengths of future segments encoded into the embedding. At the extreme of $k \to 0$, HPL theoretically degenerates to MR as the conditional reward $r_\psi$ contains no information about the future.

Figure 11 illustrates the performance of HPL across various values of $k$ for all tasks in Meta-World. As $k$ increases from zero, the performance generally improves, supporting the efficacy of future conditioning. However, beyond a certain threshold, further increases in $k$ lead to performance declines and fluctuations. This phenomenon may be attributed to the incapability of modeling excessively long trajectory segments with the VAE structure.

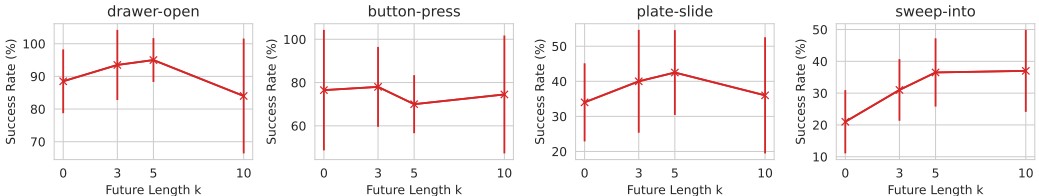

Figure 11: Success rate of HPL with various encoded future lengths $k$. We report the average and the standard deviation of the performances across 20 evaluation episodes and 10 seeds.

### E.4 Scaling with Dataset Sizes

In this section, we investigate the performance of HPL as well as MR with various dataset sizes. We vary the sizes of the preference dataset $\mathcal{D}_\mathrm{p}$ and the unlabeled dataset $\mathcal{D}_\mathrm{u}$, and plot the curve of the performances in Figure 12 and Figure 13, respectively. While both methods exhibit an upward trend in success rates as the dataset sizes $|\mathcal{D}_\mathrm{p}|$ and $|\mathcal{D}_\mathrm{u}|$ grow, HPL outperforms MR in almost every configuration. These experiments collectively confirm the superiority and scalability of HPL. In some tasks (e.g. drawer-open and sweep-into), we observe that the performance may drop as the sizes of both datasets increase. We have identified that this phenomenon is attributable to the non-uniform

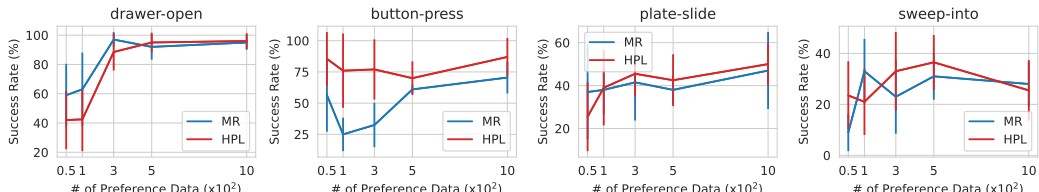

Figure 12: Success rate of HPL and MR with various sizes of $\mathcal{D}_\mathrm{p}$. We report the average and the standard deviation of the performances across 20 evaluation episodes and 10 seeds.

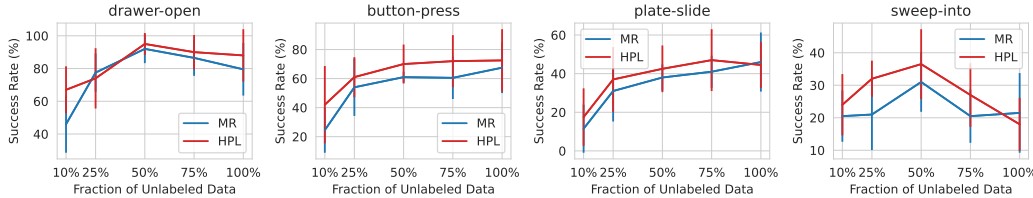

Figure 13: Success rate of HPL and MR with various sizes of $\mathcal{D}_\mathrm{u}$. We report the average and the standard deviation of the performances across 20 evaluation episodes and 10 seeds.

distribution of the datasets. In certain instances, the preceding proportion of trajectories yields greater rewards, leading to a more effective policy compared to using the whole dataset. This improvement occurs because the AWR objective used by IQL is influenced by the quality of the behavior policy.

