# OpenReview forum: "Hindsight Preference Learning for Offline Preference-based Reinforcement Learning"
_ICLR.cc/2025/Conference — Submitted to ICLR 2025_

### Official Review · Reviewer_941Q · 2024-10-30

**Soundness:** 2
**Presentation:** 3
**Contribution:** 3
**Rating:** 5
**Confidence:** 4

**Summary:**

This paper introduces Hindsight Preference Learning (HPL), a novel method for offline PbRL. It addresses the shortcomings of existing methods that rely on Markovian reward assumptions, by incorporating future trajectory outcomes in reward learning. HPL uses a variational auto-encoder (VAE) to encode future information and improves credit assignment by considering the future consequences of actions. The authors provide extensive empirical results across a variety of tasks (locomotion and manipulation) and demonstrate that HPL outperforms several state-of-the-art PbRL methods.

**Strengths:**

1. The paper is clearly written, easy to understand, and provides detailed, well-functioning code.
2. The proposed VAE method incorporates hindsight future information from trajectories into the reward model, which effectively mitigates the reward learning issue caused by the distributional shift between preference data $D_p$ and offline data $D_u$.
3. The experimental performance is impressive, and the ablation studies are thorough.

**Weaknesses:**

1. The motivation example might be unreasonable: As far as I know, in the same task, $D_u$ is often sampled from $D_p$, so they follow the same marginal distribution. When such an extreme example occurs, it indicates a significant distribution difference between $D_u$ and $D_p$. Although Section 5.3 demonstrates the effectiveness of HPL under distributional shift, this example may lack generality.
2. In fact, this method can be viewed as leveraging the unlabeled dataset to improve the robustness of the reward model. The idea is somewhat similar to previous work [1], which learns the reward model from labeled data and then labels the unlabeled data to enhance offline RL performance. A further theoretical analysis, such as exploring the theoretical performance bounds of learning the hidden dynamics from the unlabeled data in the reward model, would enrich the paper (though this could be very challenging, so it's just a suggestion).
3. In Table 7, the performance of OPPO and CPL is quite poor. Is this mainly because the preference data $D_p$ is too small?
4. In Figure 6c, reducing the size of $D_p$ leads to worse performance. Why is that?
5. The baselines and ablations lack verification of the core idea of the paper: If the authors aim to verify that the performance improvement comes from the richer environment dynamics provided by the unlabeled data, they should compare with an ablation version where the VAE learns future representations from $D_p$ itself rather than from $D_u$, to further clarify the source of the performance gain. Additionally, in Table 1, a bi-directional transformer version of PT could be used to model future information as another baseline for comparison.
[1] The Provable Benefits of Unsupervised Data Sharing for Offline Reinforcement Learning. ICLR23.

**Questions:**

Please see the weaknesses.

---

> ### Author Response · Authors · 2024-11-22
>
> Thank you for your valuable evaluations and suggestions. Below we provide further explanations for the concerns and questions about our paper.
>
> - **W1: The distribution shift between $\mathcal{D}\_{\rm u}$ and $\mathcal{D}\_{\rm p}$ may lack generality.**
>
>     We would like to note that, although in some cases $\mathcal{D}\_{\rm p}$ is sampled from $\mathcal{D}\_{\rm u}$, their magnitudes are very different since crowdsourced annotation can be expensive. In our experiments, even if the $\mathcal{D}\_{\rm p}$ is sampled from $\mathcal{D}\_{\rm u}$, the amount of data in $\mathcal{D}\_{\rm p}$ is <10% of $\mathcal{D}\_{\rm u}$, making $\mathcal{D}\_{\rm p}$ a possibly biased subset of $\mathcal{D}\_{\rm u}$. Besides, in PbRL the queries in $\mathcal{D}\_{\rm p}$ are often actively selected based on some heuristics to maximally gather information with less budget (see [1] as an example). Such non-uniform sampling inherently causes a mismatch between the datasets.
>
>     In addition, the preference distribution shift has also been identified in other works. For example, [2] discovers that the misalignment between queries and the current policy can hinder learning efficiency, and they propose to sample preference data (for human labeling) from the most recent experiences collected by the policy. At a larger and more practical scale, in LLM RLHF fine-tuning, it was found that the reward model can quickly degrade if not exposed to the sample distribution of updated language models, thus necessitating an iterative data collection procedure [3]. These findings collectively demonstrate the effect of preference shifts.
>
>     [1] Daniel Shin, et al. "Benchmarks and Algorithms for Offline Preference-Based Reward Learning."
>
>     [2] Xiao Hu, et al. "Query-policy misalignment in preference-based reinforcement learning."
>
>     [3] Hugo Touvron, et al. "Llama 2: Open Foundation and Fine-Tuned Chat Models"
>
> - **W2: Theoretical analysis.**
>
>     Thanks for your insight and recommendation! We will consider and look into the theoretical characterization of HPL.
>
> - **W3: Performance regarding OPPO and CPL.**
>
>     Yes, we believe the limited size of the preference dataset is the primary reason for the unsatisfactory performance of CPL, despite our effort to augment CPL with BC regularizations. Please refer to line 1056 for details.
>
>     For OPPO, we employed the open-sourced code by the authors without any alterations and utilized the same hyperparameters from the original paper. However, we observed significant fluctuations in the learning process and declining performances over time.
>
> - **W4: Figure 6c, increasing the size of $\mathcal{D}_{p}$ leads to worse performance, why is that?**
>
>     Figure 6 is about the analysis with the VAE, so we think you are actually referring the Figure 7c where reducing $\mathcal{D}\_{\rm p}$ instead leads to better performance for both MR and HPL.
>     The trajectories in the preference dataset are collected by three types of policy checkpoints: expert-level policy, expert policy with 1.0-std Gaussian perturbation, and random policy. Consequently, the quality and value of each trajectory can vary significantly. For $|\mathcal{D}\_{\rm p}|=100$, we conjecture that the proportion of those bad trajectories is comparatively higher than $|\mathcal{D}\_{\rm p}|=50$, which hurts the overall quality of the reward function. A similar phenomenon was also observed in the paper that provides the dataset (see Table 3 in [4]).
>
>     [4] Joey Hejna, et al. "Inverse Preference Learning: Preference-based RL without a Reward Function."
>
> - **W5.1: using $\mathcal{D}_{\rm p}$ to learn representations as a baseline**
>
>     Thanks for pointing this out. To verify this, we implemented another variant of HPL where the representation networks and the prior distribution are learned using $\mathcal{D}\_{\rm p}$. We trained for 3 seed and the performances are listed in the following table. The results for the variant that uses $\mathcal{D}\_{\rm u}$ for marginalization are from the draft.
>
>     ||hop: med-e → med-r|hop: med-r → med-e|walk: med-e → med-r|walk: med-r → med-e|
>     |:---:|:---:|:---:|:---:|:---:|
>     |representation from $\mathcal{D}_{\rm p}$|43.0±27.5|75.6±7.6|66.1±9.2|108.2±0.2|
>     |representation from $\mathcal{D}_{\rm u}$|71.1±8.3|99.5±13.8|71.8±7.7|108.4±0.6|
>
> - **W5.2: Bi-directional PT as another baseline.**
>
>     The Preference Transformer consists of a stack of causal attention layers, followed by one bidirectional attention layer at the end to calculate the importance weights of each state-action pair (see Figure 2 left in [5]). Therefore, the PT used in our experiment already incorporates future information.
>
>     [5] Changyeon Kim, et al. Preference Transformer: Modeling Human Preferences using Transformers for RL.

---

> > ### Comment · Reviewer_941Q · 2024-12-03
> >
> > I appreciate the authors' detailed response to my comments. Most of my concerns are addressed in addition to W2 and W3.

---

### Official Review · Reviewer_XGWX · 2024-10-31

**Soundness:** 4
**Presentation:** 4
**Contribution:** 3
**Rating:** 8
**Confidence:** 4

**Summary:**

This paper introduces Hindsight Preference Learning (HPL), a novel reward model learning method for offline preference-based reinforcement learning. The authors address the issue of distribution shift between preference-labeled and unlabeled datasets, which can bias the Markovian reward assumption. To tackle this problem, they propose a reward function that is conditioned not only on states and actions but also on future segments. To handle the high-dimensional nature of future segments, the authors employ a conditional VAE to transform them into embedding vectors that better integrate with the reward function. The effectiveness of the method is demonstrated through experiments on Gym-MuJoCo, Adroit, and Meta-World benchmarks, comparing against existing preference-based reinforcement learning algorithms. The results show significant improvements under preference distribution shift, while ablation studies validate the effectiveness of both the VAE and hindsight reward function.

**Strengths:**

1. The paper addresses a critical challenge in offline preference-based reinforcement learning, presenting well-motivated arguments and clearly illustrating the reward modeling issue through compelling examples.
2. The authors make the insightful observation that human preferences often stem from final outcomes. Their proposed hindsight preference learning leverages future segments to better model these preferences, resulting in a more robust reward function, particularly for datasets with outcome shifts due to multiple modalities or approaches. The integration of conditional VAE makes the algorithm practical for complex tasks like visuomotor policy learning.
3. The experimental evaluation strongly supports the core ideas, encompassing multiple popular benchmarks and competitive baselines. The comprehensive ablation studies examining future segment length, dataset sizes, and reward ensemble provide valuable insights into HPL's advantages.

**Weaknesses:**

1. The VAE implementation is limited to encoding future segments, whereas it could potentially be extended to data augmentation, prior distribution modeling, and other applications.
2. While the authors focus on "preference distribution shift" between labeled and unlabeled datasets, they assume stability in other distributions. The paper does not adequately address potential shifts in visual appearance, dynamics, and sensor noise - common challenges in real-world datasets.

**Questions:**

1. The experiments utilize a customized split of the original dataset, using medium-expert quality for the preference dataset and medium quality for the unlabeled dataset. Has there been any evaluation of preference shift on real-world, large-scale datasets?
2. Equation 9 includes the number of samples N in the reward function. How does N influence performance, and what is the practical range for this parameter?
3. The concept of utilizing future segments appears similar to graph search and retrieval-based offline RL methods [1,2]. Could HPL achieve comparable results?
[1] Deyao Zhu, Li Erran Li, and Mohamed Elhoseiny. Value memory graph: A graph-structured world model for offline reinforcement learning. arXiv preprint arXiv:2206.04384, 2022.
[2] Yin Z H, Abbeel P. Offline Imitation Learning Through Graph Search and Retrieval. arXiv preprint arXiv:2407.15403, 2024.

---

> ### Author Response · Authors · 2024-11-22
>
> Thank you for spending time reviewing our paper and also for your appreciation. Below we provide further explanations and hope they can address your concerns.
>
> - **W2: Fail to address potential shifts in visual appearance, dynamics and sensor noise.**
>
>     We acknowledge that practical applications of reinforcement learning agents must address additional challenges, such as distribution shifts in visual observations, and the community has made significant efforts to tackle these issues. However, this paper primarily focuses on the specific distribution shift between the preference dataset and the unlabeled dataset. We believe that the proposed method can be combined with other techniques to collectively enhance the robustness of RL systems.
>
> - **Q1: Preference distribution shift on real-world, large-scale datasets?**
>
>   The preference distribution shift problem has also been identified in other studies. For example, [1] discovers that the misalignment between queries and the current policy can hinder learning efficiency, and they propose to sample preference data (for human labeling) from the most recent experiences collected by the policy. At a larger and more practical scale, in LLM RLHF fine-tuning, it was found that the reward model can quickly degrade if not exposed to the sample distribution of updated language models, thus necessitating an iterative data collection procedure [2]. These findings collectively demonstrate the effect of preference shifts on real-world and large-scale scenarios.
>
> [1] Xiao Hu, et al. "Query-policy misalignment in preference-based reinforcement learning."
>
> [2] Hugo Touvron, et al. "Llama 2: Open Foundation and Fine-Tuned Chat Models."
>
> - **Q2: Effect of $N$**
>
>     We carried out experiments using the MuJoCo-Gym tasks to ablate the effect of $N$, and found that generally HPL is not sensitive to this value:
>
>     ||hop-med-rep|hop-med-exp|walk-med-rep|walk-med-exp|
>     |:---:|:---:|:---:|:---:|:---:|
>     |N=1|79.91±9.29|107.99±4.40|60.40±3.78|105.78±3.28|
>     |N=5|86.62±4.80|100.61±0.98|64.71±4.59|104.26±3.00|
>     |N=20|78.89±7.48|103.78±6.65|60.82±3.46|107.93±1.77|
>     |N=50|81.74±8.08|105.05±3.03|64.42±6.87|108.93±0.51|
>
>     It is also worth noting that the sampling cost is amortized and negligible, as we only need to label the rewards once before the reinforcement learning (RL) training begins.
>
> - **Q3: Similarity to retrieval-based offline RL methods.**
>
>     Thanks for pointing this out. The idea of leveraging trajectory segments and learning compact representations for them indeed shares similarities in methodologies with retrieval-based offline RL methods. The recommended papers both build a graph where the original states in the MDP are categorized into vertices in the graph based on learned metrics or representations, and the vertices are connected via the actions. With such a graph, [1] performs value iteration on this graph-based MDP, while [2] retrieves neighbors and trains the imitation policy using weighted behavior cloning. Nevertheless, HPL focuses on leveraging future segments for improved reward learning, rather than retrieving memory for imitation. We will add one section to discuss the similarities and connections between HPL and other demonstration-involved RL methods (e.g. retrieval-based RL, imitation RL) in the revised version.

---

### Official Review · Reviewer_QfZv · 2024-10-31

**Soundness:** 3
**Presentation:** 2
**Contribution:** 2
**Rating:** 5
**Confidence:** 3

**Summary:**

The authors study the problem of offline preference-based RL. One major problem with existing SOTA approaches, as cited by the authors, is the evaluation of trajectory segments from a global perspective, making the obtained reward implicitly dependent on the future part of the segment. The contributions of the paper can be summarized as follows:
- The authors propose HPL, which models human preferences using a reward function conditioned on the state, action, and future trajectory, referred to as hindsight information.
- HPL leverages the trajectory distribution of the unlabeled dataset to learn a prior over future outcomes, providing robust and advantageous rewards during labeling.

**Strengths:**

- The problem to be solved is clearly specified, and although the approach is simple, it effectively addresses aspects that previous papers have overlooked.
- In the experiments, HPL outperforms the existing SOTA algorithms on most datasets.

**Weaknesses:**

- I understand that the use of a gambling MDP in Section 3.1 highlights the importance of this study. However, given the very short trajectory length (two steps) in the example, it remains unclear whether hindsight information was effectively applied to achieve accurate credit assignment using future trajectory information.
- Figure 10 shows that HPL is already 2 to 10 times slower than other models. There seem to be several reasons for this slower speed.
     - Equation 8 requires summing all steps of each trajectory, which likely takes considerable time.
     - A significant amount of time is spent on sampling in Equation 9.
     - If the running time includes VAE modeling time, it is expected to take even longer.
- The core idea of HPL seems to lie not in using future trajectory to train the model, but rather in the learned VAE that represents the future trajectory. As the authors mentioned, when the future length exceeds a certain threshold, it becomes difficult to accurately represent longer future segments, which lowers HPL's performance. If future information is available, the performance should ideally be more accurate. How might this limitation be addressed?

Minor comments:

- typo on line 403: ''$\hat{P}(s_{good}|s_{1},a_{1})$ and $\hat{P}(s_{good}|s_{1},a_{1})$" -> "$\hat{P}(s_{good}|s_{1},a_{1})$ and $\hat{P}(s_{bad}|s_{1},a_{1})$"

**Questions:**

In addition to questions about potential weaknesses, I would like to raise a few more questions:
- When computing the reward using the prior distribution, how does the effect depend on the number of samples? (N=20 in appendix)
- What about using both preference data and unlabeled data when training the VAE? Wouldn't this improve the representation across multiple points?

---

> ### Author Response · Authors · 2024-11-22
>
> Thank you for your time in reviewing and catching the typo. Below we provide further explanations and experiment results for your reference.
>
> - **W1: It remains unclear whether hindsight information was effectively applied to achieve accurate credit assignment.**
>
>     Following existing practices in PbRL, we carried out Pearson analysis with the reward model obtained by MR and HPL. To be specific, we used datasets from Gym-MuJoCo tasks, and labeled the preference according to the trajectory return and used HPL/MR to learn the reward model. HPL additionally uses the vast unlabeled dataset to learn representations and the prior distribution. Afterward, we used the learned reward models to predict the rewards of data points in the unlabeled dataset and calculate the Pearson correlation coefficient between the predictions and the ground-truth rewards. The results are listed in the Table:
>
>     ||HPL Pearson|MR Pearson|HPL Performance|MR Performance|
>     |---:|:---:|:---:|:---:|:---:|
>     |hop-med-rep|0.85|0.81|80.17±13.61|48.27±14.53|
>     |hop-med-exp|0.55|0.49|103.79±6.03|83.93±37.08|
>     |walk-med-rep|0.81|0.81|82.86±1.40|73.66±4.52|
>     |walk-med-exp|0.52|0.46|110.72±0.55|102.53±13.59|
>
>     We observed that the rewards provided by HPL are more consistent with the ground-truth rewards, and this enhanced consistency translates into improved performance. These findings further highlight the effectiveness of HPL.
>
> - **W2: About the training speed.**
>
>     Yes, HPL is slower than other methods. We would like to note that the training time shown in Figure 10 includes the training of VAE, which takes up most of the time for reward model training. In comparison, the time spent on summing all steps of each trajectory and sampling is negligible.
>
> - **W3: How can the limitation about length $k$ be addressed**
>
>     The VAE is used to generate a compact representation for the future segments to facilitate sampling and training. This motivation also echoes some other literature, e.g. Hindsight Information Matching (HIM) [1], which employs a Decision Transformer-like pipeline to obtain the representations and demonstrates good performance in both offline RL and PbRL tasks. Therefore, techniques like HIM may be used to learn representations of longer sequences. Additionally, domain-specific knowledge can inspire more effective ways to represent future segments. For instance, in navigation tasks, the intended goal state could be used as a condition, offering a more tailored and effective variant of HPL for such scenarios.
>
>     [1] Hiroki Furuta, et al. Generalized Decision Transformer for Offline Hindsight Information Matching.
>
> - **Q1: Effect of $N$.**
>   We carried out experiments using the MuJoCo-Gym tasks to ablate the effect of $N$, and found that generally HPL is not sensitive to this value:
>
>     ||hop-med-rep|hop-med-exp|walk-med-rep|walk-med-exp|
>     |---:|:---:|:---:|:---:|:---:|
>     |N=1|79.91±9.29|107.99±4.40|60.40±3.78|105.78±3.28|
>     |N=5|86.62±4.80|100.61±0.98|64.71±4.59|104.26±3.00|
>     |N=20|78.89±7.48|103.78±6.65|60.82±3.46|107.93±1.77|
>     |N=50|81.74±8.08|105.05±3.03|64.42±6.87|108.93±0.51|
>
>     It is also worth noting that the sampling cost is amortized and negligible, as we only need to label the rewards once before the reinforcement learning (RL) training begins.
>
> - **Q2: Using both preference data and unlabeled data to train VAE?**
>   For most of our experiments (except for the mismatch ones), the preference dataset is sampled from the offline dataset (1% ~ 10%), and therefore there is no need to mix them. Besides, we hope the data distribution to resemble the one used for RL training, and thus we decided to use the unlabeled dataset.

---

> > ### Comment · Reviewer_QfZv · 2024-11-25
> >
> > I appreciate the authors' detailed response to my comments. However, I still have a few remaining concerns.
> >
> > - In the comment of W1, regarding the Walker2d-Medium-Replay dataset, the Pearson correlation coefficient of HPL is equal to that of MR Pearson. However, the performance of HPL is significantly better than that of MR. Both methods use the same algorithm, IQL, for policy optimization, which makes the performance difference somewhat puzzling.
> > - If time permits, it would be interesting to see the results in the MuJoCo environment for various segment lengths, $k=3$ or  $k=10$. This could lead to a work that makes better use of the hindsight information emphasized by the authors.

---

> > > ### Author Response · Authors · 2024-12-02
> > >
> > > Thanks for your reply and suggestions.
> > >
> > > - Regarding the Pearson coefficient, We hypothesize that it captures a general consistency between the learned reward and the ground truth. However, the final performance may deviate due to the sensitivity of the temporal difference learning process. Precisely quantifying the actual closeness between the learned reward and the ground truth is challenging, especially at the scale of the benchmarks used in our paper, as discussed in [1].
> > > - We conducted ablation studies for $k$ in the MuJoCo environment. We run 5 seeds for each configuration and each dataset, and the results are listed in the following table. Overall we found a similar trend to the tasks from MetaWorld: extending $k$ from 1 to 5 generally enhances performance until exceeding a certain threshold, after which the performance starts to fluctuate and decrease ($k=10$).
> > > | k | 1 | 3 | 5 | 10 |
> > > |---|---|---|---|---|
> > > | hop-m-r | 76±16.5 | 73.6±12.6 | 83.0±14.4 | 78.9±22.5 |
> > > | hop-m-e | 90.6±8.8 | 98.3±13.8 | 104.0±7.7 | 103.8±11.5 |
> > > | walk-m-r | 56.4±7.9 | 65.5±6 | 64.1±8.9 | 56±11.4 |
> > > | walk-m-e | 93.6±10.6 | 108.9±0.3 | 108.9±0.5 | 95±12.1 |
> > >
> > > [1] Joar Skalse, et al. STARC: A General Framework For Quantifying Differences Between Reward Functions. ICLR 2024.

---

### Official Review · Reviewer_vGKE · 2024-11-04

**Soundness:** 2
**Presentation:** 3
**Contribution:** 2
**Rating:** 3
**Confidence:** 3

**Summary:**

This paper studies preference-based reinforcement learning (PbRL). Notably, it proposes to consider human annotations from a holistic perspective; that is the annotations would include the considerations of future outcome instead of in-trajectory information only. Inspired by this mismatch, authors propose to model human preference with hindsight information via a VAE structure, which is further marginalized for rewarding labeling and following RL downstream tasks. Experiments are conducted on annotations from human and scripted teacher and advantageous performance is demonstrated.

**Strengths:**

**Originality**: The paper studies a very important problem in PbRL: the nature of human annotations, which is inherently more complex than scripted annotations. By proposing a VAE structure with marginalization techniques, this paper a novel and interesting method.


**Quality**: Experiments are conducted on existing human-annotated dataset where the proposed method shows greater performance.


**Clarity**: The paper is well written with illustrative examples, clear text and figures.


**Significance**: The studied problem: the mismatch between annotations and chosen reward models is of very importance and shall be beneficial for the community.

**Weaknesses:**

The primary concern lies in the **mismatch** between motivation and downstream methods.

The paper aims to incorporate additional future information when learning the $r_{\psi}$. However, actually, the $r_{\psi}$ that takes future information into account should be the Q-function rather than the reward function. Or more rigorously, the $r_{\psi}$ actually encompasses far more information than reward function, but it is not necessarily a Q-function. When downstream algorithms utilize the $r_{\psi}$, the $r_{\psi}$ is considered as reward function, employing a method that does not consider future information.

**Questions:**

The preference signals on the MetaWorld platform do not actually take into account future information. Why is the algorithm proposed in this paper considered superior?

---

> ### Author Response · Authors · 2024-11-22
>
> Thank you for your time in reviewing our paper. Below we provide explanations about certain concerns or questions you raised in the review, and we are willing to engage in further discussions if you have further questions.
>
> - **W1: Q functions, rather than the reward functions, should be dependent on future information. Besides, Downstream RL algorithms utilize $r_\psi$ as the reward function, employing a method that does not consider future information.**
>
>     Theoretically, yes, ideal rewards (or ground-truth rewards) should depend solely on the current state-action pair and remain independent of the data distribution. However, since the reward model is learned using preference labels derived from entire trajectories, the learning process inherently introduces a dependence on the training data distribution, as demonstrated in the gambling MDP example.
>
>     What HPL is doing is to correct the dependence on training data, by 1) making the dependence on the future explicit as conditions and 2) marginalizing the conditions using the target distribution (equation 9 in the paper). In this way, the predicted rewards are more aligned with $\mathcal{D}_{\rm u}$, leading to improved performance during downstream RL optimization. HPL does not aim to leverage future information in the RL stage, as the rewards are already corrected and aligned with the unlabeled dataset.
>
> - **Q1: The preference signals on the MetaWorld platform do not consider future information.**
>
>     Similar to W1, the preference signals are assigned by evaluating the goodness of the whole trajectory and this causes the learned reward to have an implicit dependence on the rest part of the trajectories.

---

> > ### Comment · Reviewer_vGKE · 2024-11-28
> >
> > Thank you for your response. However, my concern **W1** still remains. The primary concern is that the $r_{\phi}$ contains additional **future information** compared with reward function, yet it is still treated as a reward function when utilized, neglecting the potential impact of this extra **future information** on downstream algorithms.

---

> > > ### Author Response · Authors · 2024-12-02
> > >
> > > Thank you for your response. We would like to further clarify the points we made in our previous reply regarding W1. Regardless of the algorithm used—whether HPL or the baseline method MR—the learned $r_\psi$ inevitably depends on the training data distribution, as we emphasized in our last response to W1. The key distinction lies in how this dependence is handled: the baseline method MR introduces an implicit and biased dependence, whereas HPL explicitly models this dependence and actively corrects the mismatch.

---

### Official Review · Reviewer_r8hL · 2024-11-04

**Soundness:** 3
**Presentation:** 4
**Contribution:** 2
**Rating:** 5
**Confidence:** 3

**Summary:**

The paper introduces a new preference learning method that utilizes hindsight information. By incorporating future states into the reward through a VAE structure, HPL can generate more robust rewards and overcome the limitations of Markovian rewards.

**Strengths:**

- The paper introduces a framework for hindsight preference learning, with a practical implementation using a VAE architecture.
- I think using the example of a gambling MDP greatly helps in understanding the motivation for using hindsight preference learning.
- Experimental results support the effectiveness of HPL.

**Weaknesses:**

- Although I understand the benefits of hindsight preference learning over Markovian rewards, the authors do not clearly position their paper within the broader research on non-Markovian rewards. As mentioned in the related work, both the Preference Transformer and Hindsight PRIOR  assume non-Markovian rewards and consider the full trajectory when calculating rewards. Given this, why does the paper primarily motivate its approach as an improvement over Markovian rewards, despite the existence of several approaches that do not rely on this assumption?
- For example, what is the fundamental difference between this paper and the Hindsight PRIOR paper, as Hindsight PRIOR considers “the importance of a state in a trajectory given that a future state was reached” [1]? Please clarify the novelty of this work in comparison to other hindsight-based PbRL papers.
- The gambling example is very helpful in illustrating the motivation. However, I am concerned that in more stochastic settings, hindsight modeling may introduce noise and add excessive complexity. The ablation study on k, while focusing on a different message, seems to somewhat support this concern.

References

[1] Verma, M., & Metcalf, K. (2024). Hindsight PRIORs for Reward Learning from Human Preferences.

**Questions:**

- Why do the segment encoder q, decoder p, and prior f all use the same notation, θ, for their parameters?
- How does the distribution mismatch between Dp and Du cause problems, and how does HPL address this issue (through improved credit assignment) in the main benchmark experiments? Is there any analysis on these tasks other than the example of gambling MDP?

---

> ### Author Response · Authors · 2024-11-22
>
> Thank you for your valuable evaluations and suggestions.
>
> - **W1: Positioning HPL within the research on non-Markovian rewards.**
>
>     HPL still assumes the Markovian property, i.e. the ground truth reward only depends on the current state and action, rather than the history. Hindsight PRIOR also makes such an assumption, as it learns a reward model that takes the state and action as input (we mistakenly stated that Hindsight PRIOR uses non-Markovian rewards in the draft), while Preference Transformer employs non-Markovian rewards predicted by transformer layers.
>     We think the primary challenge that HPL hopes to address is how to attribute rewards to individual state-action pairs when preferences are based on evaluating an entire trajectory. HPL does this by first assigning credits to conditional rewards and then marginalizing the conditions. Preference Transformer and Hindsight PRIOR, on the other hand, adjust the importance weights of each state-action pair using some attention scores. It is also applicable to extend HPL and learn non-Markovian rewards, by using RNNs to encode the history into an embedding $h_t$ and feeding this embedding to the VAE networks. The ELBO objective (Eq. 7) remains unchanged, similar to how previous works (e.g. PlaNet, Dreamer) train the RSSM models.
>
> - **W2: Comparison to Hindsight PRIOR**
>
>     Hindsight PRIOR hypothesizes that the attention scores between the last state (as queries) and the current state-action pair (as keys) can measure the importance weight and redistribute the rewards. This heuristic approach may be challenged in situations where the assumption may not hold, for example, when attention to the last state is irrelevant to the preference.
>     HPL does not incorporate such heuristics into preference modeling. Instead, it considers a scenario where a trajectory is preferred not because the current state and action are good, but because the agent made rewarding decisions in subsequent steps. Future-conditional rewards can capture and separate the dependence on the future, and reflect the actual utility of the state-action pair. Similar motivations are also shared by literature on credit assignment, e.g. [1].
>
>     [1] Thomas Mesnard, et al. "Counterfactual Credit Assignment in Model-Free Reinforcement Learning."
>
>
> - **W3: Modeling difficulty in more stochastic environments.**
>
>     In more stochastic environments, the distribution of future segments can be highly complex due to the stochasticity of both the policy and the dynamics. We agree that the current approach (using VAE to compress future segments), although general, can face challenges in summarizing the future as $k$ grows. However, we note that it is possible to design more effective ways to represent future segments with domain knowledge. For instance, in navigation tasks, the intended goal state could be used as a condition, providing a more targeted and meaningful summary.
>
>
> - **Q1: Why do the encoder, the decoder and the prior share the notation of parameters?**
>
>     We use $\theta$ to denote the collections of the parameters of the encoder, decoder, and the prior. They are also jointly optimized using the ELBO loss.
>
> - **Q2: How does that distribution mismatch between the datasets cause problems? How does HPL address this issue?**
>
>     Humans often assess the desirability of a sequence of actions by considering the overall outcome, and this gives the learned reward model an implicit dependence on the training data distribution ($\mathcal{D}\_{\rm p}$). HPL mitigates this issue by making the dependence explicit as the condition and marginalizing the condition using statistics from the target distribution ($\mathcal{D}_{\rm u}$).
>
>     To demonstrate the improved credit assignment of HPL quantitatively, we followed existing practices in PbRL and carried out Pearson analysis with the reward model obtained by MR and HPL. Specifically, we used datasets from Gym-MuJoCo tasks, **labeled the preference according to the trajectory return**, and used HPL/MR to learn the reward model. HPL additionally uses the vast unlabeled dataset to learn representations and the prior distribution. We then used the learned reward models to predict the rewards for data points in the unlabeled dataset and calculated the Pearson correlation coefficient between these predictions and the ground-truth rewards. The results are listed in the Table:
>
>   ||HPL Pearson|MR Pearson|HPL Performance|MR Performance|
>   |:---:|:---:|:---:|:---:|:---:|
>   |hop-med-rep|0.85|0.81|80.17±13.61|48.27±14.53|
>   |hop-med-exp|0.55|0.49|103.79±6.03|83.93±37.08|
>   |walk-med-rep|0.81|0.81|82.86±1.40|73.66±4.52|
>   |walk-med-exp|0.52|0.46|110.72±0.55|102.53±13.59|
>
>     We observed that the rewards provided by HPL are more consistent with the ground-truth rewards, and this enhanced consistency translates into improved performance. These findings further highlight the effectiveness of HPL.

---

> ### Comment · Reviewer_r8hL · 2024-11-25
> **Response to Rebuttal**
>
> Thank you for your constructive feedback.
>
> Although some concerns have been addressed, it is still unclear how HPL would benefit. What I asked is the specific qualitative example of how it benefits in your benchmark task. Also. while it somewhat improves over heuristic attention maps, the additional computation introduced by the VAE to represent future trajectories, along with some limitations mentioned in the paper, raises concerns. Additionally, it might be helpful for the authors to directly compare the results with Hindsight Prior as a baseline.
>
> Therefore, I will maintain my score.

---

> > ### Author Response · Authors · 2024-12-02
> >
> > Thank you again for your valuable suggestions. We will continue to improve our paper based on your feedback.

---

### Meta-Review · Area_Chair_5tXX · 2024-12-19

**Metareview:**

**summary**

This paper introduces HPL, which incorporates human annotations that consider future trajectory outcomes rather than just in-trajectory information. By leveraging VAE to encode high-dimensional future segments into embedding vectors, HPL conditions the reward function on states, actions, and future outcomes, effectively addressing distribution shifts between labeled and unlabeled datasets. Empirical results across diverse tasks demonstrate that HPL significantly outperforms state-of-the-art PbRL methods, providing robust and advantageous performance under preference distribution shifts.

**strengths**

* Simple yet effective method with clear motivation
* Extensive experiments on diverse benchmarks (e.g., Gym-MuJoCo, Adroit, Meta-World) demonstrate that HPL outperforms state-of-the-art PbRL methods.
* The paper is well-written with clear text, illustrative figures, and thorough explanations.

**weaknesses**

* The distinction between HPL and existing works like Hindsight PRIOR, which also considers the importance of states based on future outcomes, is not clearly described.
* Ambiguous design choice in utilizing reward. The proposed reward function includes additional future trajectory information, but this information is treated as a standard reward function during downstream tasks. This approach may overlook the impact of this added information, potentially leading to biases or mismatches in reinforcement learning algorithms.
* The effectiveness of HPL heavily depends on the quality of future trajectory representation via the VAE. As future segments grow longer, the VAE struggles to encode them accurately, which diminishes performance. The paper lacks a clear strategy to mitigate this limitation and improve handling of long future trajectories.

**decision**

Even though the proposed idea is interesting, several design choices are ambiguous and similarity with prior works limits the originality of the work. During the discussion phase, some comments were partially addressed, but the paper requires revision to fully resolve these issues.

**Additional Comments On Reviewer Discussion:**

The author's responses on design choices and comparison with prior work were not enough to convince several reviewer's concern.

---

### Decision · Program_Chairs · 2025-01-22

Reject